# MAPLe: Masked Adapter Prototype Learning for OOD Generalization

## Abstract

Parameter-efficient fine-tuning with adapters (e.g., LoRA) equips LLMs with task-specific skills. However, utilizing multiple pretrained adapters for out-of-distribution (OOD) generalization remains challenging. Existing techniques for OOD generalization using multiple pretrained LoRAs, route inputs using LoRA representations (*prototypes*) obtained independently, assuming these representations capture complementary information. However, we observe that for existing methods, in-distribution and OOD routing entropies are often comparable, thus bringing the complementarity assumption into question. We derive the theoretical conditions that could lead to a violation of such assumptions, distilling the cause down to the presence of shared, noisy prototype subspaces. Based on this, we introduce **MAPLE (Masked-Adapter Prototype LEarning)**, a simple learning framework that refines LoRA prototypes by masking the target task's LoRA during prototype learning. In doing so, it encourages prototypes to discard noisy attributes, which improves routing and strengthens OOD generalization. Extensive experiments on language models of varying size, such as Phi-2 (2.7B) and LLaMA-3 (8B) equipped with heterogeneous pools of pretrained LoRAs, show that MAPLE improves the LoRA representation and thus achieves state-of-the-art performance across multiple benchmarks.

## 1 Introduction

Large Language Models (LLMs) have emerged as versatile, general-purpose tools with a wide range of practical applications. However, their performance can vary substantially depending on the task under consideration (Moayeri et al., 2025). To mitigate this variability, lightweight fine-tuning techniques, such as LoRA (Hu et al., 2022), have gained traction, enabling the efficient adaptation of LLMs to diverse downstream tasks when task-specific training data is available. With the growing availability of pretrained LoRA adapters, recent work studies a mixture-of-experts approach, *MoErging* (Yadav et al., 2025), which aims to generalize to Out-Of- Distribution (OOD) inputs by dynamically combining adapters from a pretrained pool. Central to these methods are *prototypes*: task-level representations used to compute routing weights for each input. Prior work constructs prototypes via heuristics (Muqeeth et al., 2024; Chronopoulou et al., 2023; Cheng et al., 2024; Durbin, 2024; Maxine, 2023) or learning-based objectives (Diao et al., 2023; Ostapenko et al., 2024), with the goal that prototype similarity reflects an adapter's functional suitability for a given input. With appropriately determined routing weights, knowledge from pretrained LoRAs can dynamically augment the base model weights to generalize to OOD inputs.

Although prototypes obtained by existing methods have been shown to facilitate effective routing, these prototypes are constructed in isolation, assuming they remain optimal when used jointly. To investigate the effectiveness of these LoRA prototypes in identifying relevant LoRAs for a given input, we compute the entropy of routing coefficients on ID tasks and compare it to that of OOD tasks. Ideally, if the LoRA representations are well discriminative, the routing entropy for In-Distribution tasks would be lower compared to OOD tasks. On the contrary, we empirically observe that the entropy of routing coefficients for in-distribution tasks is comparable to that of OOD tasks (see Fig. 1), indicating the inadequacy of LoRA prototypes in identifying relevant LoRAs effectively when used jointly. From a representation learning perspective, this can be interpreted as LoRA representations encoding certain non-discriminative/noisy attributes. We theoretically show how such representations restrict the task subspace spanned by the adapters, thereby degrading OOD generalization.

To enhance LoRA representations and thereby improve OOD generalization capabilities, we introduce a novel prototype refinement scheme: **MAPLE** (*Masked Adapter Prototype LEarning*), a training framework that improves prototypes' discriminability by simulating OOD conditions. During prototype refinement, MAPLE *masks the target task's adapter*, forcing routing to rely on informative, task-specific cues rather than generic shared/noisy features. This simple intervention yields prototypes with improved discriminability, produces lower routing entropy on ID inputs, and enables more effective composition of pretrained adapters at test time. We empirically validate the effectiveness of the proposed method on two widely used LLMs, Phi-2 (2.7B)(Javaheripi et al., 2023) and LLama-3 (8B)(Grattafiori & Dubey, 2024).

In summary, we make the following contributions.

- We identify and formalize the inadequacy of LoRA prototypes learned in isolation for MoErging: Prototypes learned in isolation exhibit dominant, noisy subspaces that influence routing plans irrespective of the input distribution (leading to similar in-distribution and OOD routing entropies), resulting in poor generalization.

- We propose MAPLE, a masked-adapter prototype learning framework that enhances prototypes by discarding noisy attributes and thereby improving OOD generalisation.

- Extensive experiments and ablations across heterogeneous benchmarks demonstrate consistent improvements over state-of-the-art routing baselines on LLMs of varying sizes with diverse LoRA collections.

## 2 RELATED WORK

MoErging techniques can be broadly classified based on their strategy for computing routing coefficients: *embedding-based*, *optimization-based*, and *learning-based*. In this section, we review representative works from each category, highlight their key differences, and contrast them with MAPLE.

**Embedding-based methods:** These approaches build fixed *expert* representations and route by their similarity to the input. Most of these approaches generate task-level embeddings by averaging features extracted from a pretrained backbone over the task data, e.g., Token-Level Adaptation (Belofsky, 2023), LoRARetriever (Zhao et al., 2024), Mo'LoRA (Maxine, 2023), and DAM (Cheng et al., 2024). While simple and efficient, such averages tend to encode generic factors from the pretrained backbone, yielding weak separation among related adapters and limiting fine-grained routing. To reduce dependence on pretrained models, Ostapenko et al. (2024) derive embeddings directly from LoRA weights via singular value decomposition (SVD). However, because these embeddings are typically constructed *per expert*, they capture each expert in isolation and fail to account for relationships between experts; this can reduce discriminability when multiple experts are used jointly, making it harder for the router to select or combine the most appropriate experts. MAPLE counters these limitations by learning embeddings under simulated OOD conditions that discourage generic shared components and enhance cross-expert separability, resulting in more discriminative, task-specific LoRA prototypes.

**Optimization-based methods:** This class of approaches, like LoRAHub (Huang et al., 2024), MoLE (Wu et al., 2024), and Lora-Flow (Wang et al., 2024), assumes access to the target task and directly optimizes routing coefficients, either via gradient descent or gradient-free techniques. These methods are inherently task-specific and can capture intra-expert relationships while filtering out redundant information that does not contribute to routing. However, their primary goal is to maximize performance on a single, well-defined task, rather than to produce adaptable representations. By contrast, MAPLE aims to learn expert embeddings that generalize to unseen tasks in a zero-shot manner. Consequently, MAPLE and optimization-based approaches pursue largely orthogonal objectives(generalization in the presence and absence of target data), rendering direct comparisons between them less meaningful.

**Learning-based methods:** Learning-based methods train a router or gating mechanism to minimize loss on in-distribution data. For instance, PHATGOOSE (Muqeeth et al., 2024) and Co-LLM (Shen et al., 2024) learn per-expert sigmoid gates using a masked language modeling loss to decide whether to activate each expert, and apply these gates jointly during inference to control token-level

routing. MeteoRA (Xu et al., 2025) optimizes the router using expert training data to enable automatic task switching. While effective for in-distribution tasks, these methods are not explicitly designed for out-of-distribution generalization. By contrast, MAPLE emphasizes the creation of expert embeddings that remain discriminative and adaptable across diverse tasks, addressing a key limitation of existing learning-based approaches.

Embedding-based methods are simple but prone to generic prototypes; optimization-based methods excel when target data is available, thus do not work in zero-shot settings; learning-based routers fit in-distribution data but under-specify OOD discriminability. MAPLE complements all three by learning embeddings tailored to routing for OOD inputs.

## 3 METHODOLOGY

In this section, we first examine the failure modes of two representative routing methods using *routing entropy* as a diagnostic metric. We then provide a theoretical analysis to elucidate the underlying causes of these behaviors. Building on these insights, we introduce a simple yet effective training paradigm, **MAPLE**, designed to improve the discriminability and adaptability of LoRA representations for OOD generalization.

### 3.1 MOTIVATION

As discussed in section 2, most existing routing schemes construct prototypes independently, which are then used jointly during inference. We investigate whether such prototypes that are learned in isolation remain discriminative under joint use. We assess the discriminability of LoRA prototypes via *routing entropy*, an information-theoretic measure of the router's uncertainty over experts. The choice of this metric is motivated by its strong correlation with prediction loss as well as its use as a proxy objective for test-time estimation of routing coefficients (Yang et al., 2024). Given the routing distribution $p(e \mid x)$ over $M$ experts, we define

$$H(x) = -\sum_{e=1}^{M} p(e \mid x) \log p(e \mid x). \tag{1}$$

We compute $H(x)$ at the token level and report the average over tokens and samples.
Lower $H$ indicates a concentrated probability mass and, hence, a more decisive expert selection. Ideally, for in-distribution (ID) tasks (*i.e.*, LoRA trained on the input task exists in the LoRA pool), the routing entropy should be lower. While for out-of-distribution (OOD) tasks, the routing entropy is expected to be higher, reflecting greater uncertainty. Routing entropy is computed for both ID and OOD tasks using prototypes obtained from high-performing embedding and learning-based methods. As shown in Fig. 1, across five ID/OOD tasks, the routing entropy estimated with Arrow (Ostapenko et al., 2024) and PHAT-GOOSE (Muqeeth et al., 2024) prototypes does not differ appreciably between ID and OOD samples. The overlapping routing entropy distribution for ID and OOD samples suggests the insufficiency of these representations in effectively identifying relevant LoRAs under joint use. This could be due to representations en-

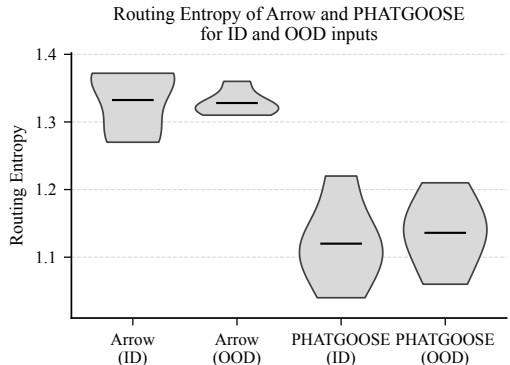

Figure 1: Distribution of routing entropy for Arrow and PHATGOOSE across five ID/OOD task pairs. ID and OOD distributions substantially overlap, indicating limited discriminability of prototypes under joint use.

coding certain noisy attributes, which blur inter-adapter distinctions and thus yield high routing uncertainty.

## 3.2 THEORETICAL FORMULATION

We start by formalizing our argument that for a set of experts to generalize OOD, it must be accompanied by an increased entropy, relative to ID samples, when applied to OOD samples. We then derive the conditions under which ID and OOD routing entropies might be similar, as observed by us in the default setting. As a remedy, we show that the desired ID-OOD entropy discrepancy can be achieved via a denoising mechanism that removes a specific type of subspace from the basis induced by the LoRA prototypes. To this end, we prove the following key results:

1. Theorem 1, where we introduce the idea of $\delta$-restrictivity, which implies that higher the difference in the ID and OOD routing entropies, *i.e.*, low ID and high OOD routing entropies, the better the OOD generalization.

2. Theorem 2, which establishes that if a router basis has a dominant subspace $\eta$ oriented disproportionately along specific distributions (*i.e.*, high $\delta$-restrictivity from Theorem 1), then, such a subspace, on average, is more noisy across the space of all distributions.

Therefore, better OOD generalization can be achieved by denoising the basis induced by the LoRA prototypes through the removal of irrelevant subspaces $\eta$ such that $\|\eta x\|$ is low on average across the space of all distributions. We defer all proofs to Appendix A.

**Definition 1** (Completeness). We call a function class of routers $\mathcal{F}$ acting on a data distribution $\mathcal{D}$ to be equipped with a complete basis $\mathcal{B}$ (corresponding to the set of LoRA prototypes) *iff*:

$$\forall (x, y) \in \mathcal{D}, \exists f \in \mathcal{F}, f(x) \in \text{span}(\mathcal{B}) \mid g_{\theta + \phi(f(x))}(x) = y,$$

*i.e.*, $\mathcal{B}$ corresponds to the feature space under which the labels $y$ of all points $x \in \mathcal{D}$ are predictable, where $g$ is a non-linear model parameterized by weights $\theta$, LoRA parameters $\phi$ and router $f(x)$.

**Out-of-Distribution Risk:** Consider the space of data distributions $\mathcal{D} = \{S\} \cup \mathcal{T}$, where $\mathcal{T}$ is arbitrarily discretized as $\mathcal{T} = \{T_1, T_2, ..., T_t\}$. The risk of a predictor $f_S$ trained on the source distribution $S$ to generalise out-of-distribution to the target space $\mathcal{T}$ is given as follows:

$$R_{\mathcal{T}}(f_S) = \frac{1}{|\mathcal{T}|} \sum_{T \in \mathcal{T}} \mathbb{E}_{(X,Y) \sim T}[\mathcal{L}(f_S(X), Y)].$$

where $(X, Y) \in T$ are samples in a given target domain $T \in \mathcal{T}$.

**Routing Plan:** The routing plan corresponding to a router $f$ with an associated basis $\mathcal{B}$ is a probability distribution $P_{\mathcal{B}}(X)$ over the basis vectors in $\mathcal{B}$ such that the mass on the $i$-th basis vector $w_i$ is given by:

$$P_{\mathcal{B}}(X)\{w_i\} = \int_{x \in X} \frac{w_i \cdot x}{\sum_{j=1}^n w_j \cdot x} \, p(x) \, dx,$$

where $X$ is some input distribution and $p(\cdot)$ is its probability density function.

### 3.2.1 CONVERGENCE STATES

Below, we enumerate the states to which the optimization over $\mathcal{F}$ might converge in terms of the relationship of the basis associated to the solution $f$ with the input distributions.

**(State 1)** $R_S(f) > K, R_{\mathcal{T}}(f) > K; R_{\mathcal{T}}(f) - R_S(f) < \epsilon$: High source and target risks, all basis vectors are irrelevant. Corresponds to a high entropy routing plan for both ID and OOD samples.

**(State 2)** $R_S(f) < k, R_{\mathcal{T}}(f) > K; R_{\mathcal{T}}(f) - R_S(f) > \gamma$: Low source but high target risk, basis vectors are aligned to the source distribution. Routing plan has low entropy for the source but high entropy for the target.

**(State 3)** $R_S(f) > K, R_{\mathcal{T}}(f) < k; R_S(f) - R_{\mathcal{T}}(f) > \gamma$: High source but low target risk, highly unlikely, not possible with ERM, only possible if $f$ is sampled randomly from the associated hypothesis class. Routing plan has high entropy for the source but low entropy for the target.

**(State 4)** $R_S(f) < k, R_{\mathcal{T}}(f) < k; R_{\mathcal{T}}(f) - R_S(f) < \epsilon$: Low source and target risks, relevant orthogonal basis, best OOD generalization among all. The routing plan has low entropy for the source but high entropy for the target.

$R_S$ above is simply the source/training distribution risk. Also, $K, k, \gamma, \epsilon$ are constants such that $K > k$ and $\gamma > \epsilon$. Given a model with high source and target risk and a high entropy routing plan, *i.e.*, state 1, the objective is to transition to a basis in state 4. However, note that the same ratios of relative entropy characterize the routing plans for both state 2 and state 4 and are only distinguishable by relative risk, the latter being determined by the level of noise in the solution. Thus, although ensuring that the routing plan induces a low source and high target entropy is a necessary condition for generalization (shown via Theorem 1), sufficiency can only be achieved with a basis that is free from noisy subspaces (Theorem 2). In the following section, we expound the scenario where despite the associated function class $\mathcal{F}$ being equipped with a complete basis (Definition 1), certain members of $\mathcal{F}$ might get restricted to a feature domain where a noisy subspace dominates, the elimination of which could lead to better generalization.

### 3.2.2 PSEUDO-INCOMPLETENESS

**Theorem 1** (Restrictivity). *Let $\mathcal{B}$ be a matrix of basis vectors associated with router $f$. Let $\eta$ be a subspace that is removed from $\mathcal{B}$, resulting in $\mathcal{B}' = \mathcal{B} - \eta$. Let $X^i$ and $X^o$ respectively be souce (ID) and target (OOD) distributions. Let the restrictivity ratio of $\eta$ be defined by:*

$$r(\eta) = \frac{H(P_{\mathcal{B}'}(X^o)) - H(P_{\mathcal{B}'}(X^i))}{H(P_{\mathcal{B}}(X^o)) - H(P_{\mathcal{B}}(X^i))} \leq \delta,$$

*where $P_{\mathcal{B}}(X)$ and $P_{\mathcal{B}'}(X)$ represent the corresponding routing plans, $H(\cdot)$ is their entropy, and $\delta$ is some constant, such that the subspace $\eta$ is called $\delta$-restrictive, i.e., the higher the value of $\delta$, the more restrictive the effect of $\eta$ on $\mathcal{B}$. Then, for subspaces $\eta_1, \eta_2 \in \mathcal{B}$, the following holds:*

$$\texttt{erank}(B - \eta_1) > \texttt{erank}(B - \eta_2) \implies r(\eta_1) > r(\eta_2),$$

*where $\texttt{erank}$ denotes the effective rank (Roy & Vetterli, 2007), i.e., the higher the effective rank of $\mathcal{B} - \eta$, the more restrictive the subspace $\eta$.*

As a result, the presence of subspaces with high $\delta$ in $\mathcal{B}$ renders the latter pseudo-incomplete for the associated $f \in \mathcal{F}$. If ID and OOD routing distributions are roughly similar, it implies the existence of a dominant subspace that is equally similar or dissimilar to both ID and OOD samples. If the subspace is strongly relevant to both ID and OOD samples, then its removal should mean that $\mathcal{B} - \eta$ would contain irrelevant basis vectors for both ID and OOD samples, leading to high entropies that are still roughly similar across both ID and OOD. In this case $X^o \cdot \eta \approx X^i \cdot \eta > \gamma$, *i.e.*, $\eta$ is very similar to both ID and OOD samples. On the other hand, and upon the removal of $\eta$, for all ID assignments, if the ID distributions lose entropy, then it implies that $\mathcal{B} \setminus \eta$ is closer to the canonical basis of distributions than $\mathcal{B}$, implying that $\eta$ was noisy. In this case $X^o \cdot \eta \approx X^i \cdot \eta < \epsilon$, where $\gamma >> \epsilon$, *i.e.*, $\eta$ is highly disssimilar to both ID and OOD samples.

**Theorem 2** (Noisiness). *The fraction of subspaces in $\mathcal{D}$ for which $\eta$ is noisy, i.e., the noisiness of $\eta$ wrt $\mathcal{D}$, is inversely proportional to the corresponding restrictivity coefficient $\delta$.*

According to the No Free Lunch theorem Wolpert & Macready (1997), any basis that induces a non-uniform distribution for an input is suboptimal on average to OOD samples. Thus, a uniform distribution corresponds to the best generalization on average across all distributions. However, in practice, some distributions are more likely to occur than others. Thus, the optimal choice in practice is to establish a middle ground with the requisite level of uniformity. The point of the theorem is to show that if the elimination of a subspace leads to an increase in entropy / more uniformity, then that elimination leads to better OOD generalization, implying that the eliminated subspace was noisy.

Theorem 1 shows that comparable ID and OOD entropies can arise from a dominant subspace that is similarly aligned with both ID and OOD samples, while Theorem 2 establishes that such subspaces degrade OOD generalization proportional to their degree of noisiness. In the next section, we present a simple method to remove such noisy components and thereby improve OOD generalization.

### 3.3 MAPLE: MASKED ADAPTER PROTOTYPE LEARNING

Theorems 1 and 2 show that noisy components in the LoRA prototypes have a restrictive effect on the routing distributions, hindering OOD generalization. If we could attenuate these noisy components within the LoRA prototypes, routing would improve, and so would OOD performance. A natural

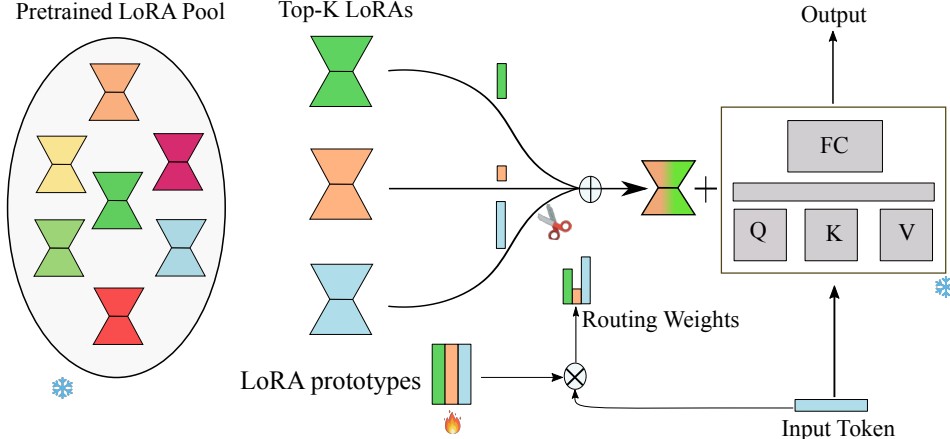

Figure 2: **MAPLE overview:** MAPLE improves OOD generalization of a pretrained LLM using a library of pretrained LoRA adapters. For each input, per-layer routing weights are computed as dot products between hidden states and learned prototypes. From the top-$k$ candidates, the LoRA associated with the input's source domain is masked out (if present). The remaining LoRAs are linearly combined according to the routing weights and used to augment the base model.

idea is to jointly train all LoRA prototypes so that noisy components contribute to higher loss and are therefore eliminated; in practice, however, joint training yields little to no improvement as shown in Table 2. We hypothesize that this failure stems from including the target-task LoRA in the routing during training. Because the target LoRA alone can substantially reduce the loss, causing gradients that would otherwise pressure non-target prototypes to unlearn their noise to become very small.

To address this and to facilitate efficient removal of noisy attributes from LoRA prototypes, we propose **Masked Adapter Prototype Learning (MAPLE)**, a training paradigm that explicitly excludes the target-task LoRA from the forward pass during training. By masking the target adapter, the model must route through the remaining prototypes, increasing the training loss and producing stronger, more informative gradients on those prototypes. This promotes the removal of spurious, task-specific noise from their representations, leading to cleaner prototypes, more reliable routing, and improved OOD generalisation. For each input, MAPLE computes routing scores in every layer, selects the top-$k$ candidate LoRAs, and explicitly *masks* the adapter trained on the input's target task. The remaining adapters are then combined using their routing coefficients to augment the base LLM. This procedure is applied layer-wise, and training proceeds with a masked language modelling objective. A schematic of the MAPLE training paradigm is shown in figure 2 and a detailed pseudocode is given in algorithm 1. At evaluation time, all adapters are used for computing routing coefficients, and routing is performed on top-$k$ LoRAs.

## 4 EXPERIMENTAL SETUP

We evaluate the effectiveness of MAPLE on two different LLMs of varying sizes and compare against competitive baselines on standard benchmarks. This section details the models, adapter pools, baselines, and benchmarks used for empirical validation. Detailed implementation specifics and hyperparameters necessary for reproducibility are provided in Appendix B.

**Models, Data and LoRA pools:** We consider two LLMs of varying size: **Phi-2** (2.7B; Javaheripi et al. 2023) and **Llama 3** (8B; Grattafiori & Dubey 2024). For Phi-2, we adopt the 256-adapter pool used in (Ostapenko et al., 2024), where each LoRA is obtained by fine-tuning on one of 256 FLAN-V2 tasks (Longpre et al., 2023). For Llama 3, we follow Xu et al. (2025) and use a 28-adapter pool trained on distinct tasks. When pretrained checkpoints are available, we reuse them; otherwise, we train the adapters using the hyperparameters and data splits provided by the authors. During MAPLE training, we update only the LoRA prototypes and thus improvements arise solely from improved routing. Additionally, for larger adaptor pools, such as the 256-adapter pool used with

---

**Algorithm 1:** Pseudocode for MAPLE

---

**Input:** Collection of pretrained LoRAs $\mathcal{A} = \{A_1, \ldots, A_L\}$; routing parameters $\{W_\ell \in \mathbb{R}^{L \times d}\}_{\ell=1}^N$,
      where $N$ is the number of layers; Task specific data $\mathcal{D} = \{\mathcal{D}_1, \ldots, \mathcal{D}_L\}$; $K$-Number of LoRAs to
      select

**Output:** Updated routing parameters $\{W_\ell\}$

```
// Notation:  h_ℓ ∈ ℝ^d is the hidden state at layer ℓ with dimension d
```

**while** *training* **do**
    Sample task $k \sim \text{Unif}\{1, \ldots, L\}$; sample $x \sim \mathcal{D}_k$;
    $h_0 \leftarrow \text{ENCODE}(\text{TOKENIZE}(x))$;
    **for** $\ell \leftarrow 1$ **to** $N$ **do**
        $z \leftarrow W_\ell\, h_{\ell-1} \in \mathbb{R}^L$;                    `// z_i scores adapter A_i`
        $\mathcal{I} \leftarrow \text{top-K}(z, K)$;
        $p \leftarrow -\infty \cdot \mathbf{1}_L$;  $p_\mathcal{I} \leftarrow z_\mathcal{I}$;           `// top-K selection`
        $\alpha \leftarrow \text{softmax}(p)$;
        $h_\ell \leftarrow \text{LLMFORWARD}\big(\theta, \{A_i\}_{i=1}^L, \alpha\big)(h_{\ell-1})$;
    $\mathcal{L} \leftarrow \text{MLM\_Loss}(h_N, x)$;
    Update $\{W_\ell\}_{\ell=1}^N$ by backprop on $\mathcal{L}$;       `// frozen base LLM and LoRAs 𝒜`

---

Phi-2, we require only a small amount of task-specific data ($\sim$200 samples per task). This high data efficiency is possible due to the large diversity of LoRA, resulting in diverse gradients, which makes it easier to eliminate noisy attributes.

**Baselines:** To evaluate the effectiveness of MAPLE in eliminating generic (noisy) factors encapsulated in prototypes and improving routing, we refine LoRA prototypes obtained using two popular and representative routing methods: ARROW (embedding-based; Ostapenko et al., 2024) and PHAT-GOOSE (learning-based; Muqeeth et al., 2024). We also report results of **Base LLM** (no adapters), **Uniform** (equal routing weights over all $K$ adapters, i.e., $p_k = 1/K$), and **Shared** (a single multi-task LoRA jointly fine-tuned on all training tasks) for comparison.

**Benchmarks:** We assess effectiveness on ten widely used tasks spanning four categories: *(1) Common-sense reasoning*—WinoGrande (Sakaguchi et al., 2019), HellaSwag (Zellers et al., 2019), PIQA (Bisk et al., 2020); *(2) Question answering*—BoolQ (Clark et al., 2019), OpenBookQA (Mihaylov et al., 2018), ARC-Easy and ARC-Challenge (Clark et al., 2018); *(3) Coding*—HumanEval (Chen et al., 2021), MBPP (Austin et al., 2021); *(4) General-purpose reasoning*—BBH (Suzgun et al., 2023). We report the zero-shot accuracy on each of these benchmarks, along with average accuracy.

| Model | Method | ARC-C | ARC-E | BBH | BoolQ | HSWAG | HE | MBPP | OQA | PIQA | WG | Mean |
|---|---|---|---|---|---|---|---|---|---|---|---|---|
| PHI-2 (2.8B) | Base | 52.90 | 77.50 | 48.00 | 82.70 | 72.50 | 45.10 | 56.00 | 49.80 | 79.20 | 75.70 | 64.15 |
| | Shared | 55.80 | 83.20 | 48.40 | 82.40 | 73.40 | 46.30 | 58.40 | 50.40 | 80.40 | 76.60 | 65.50 |
| | Uniform | 53.90 | 78.70 | 47.90 | 83.30 | 73.10 | 44.50 | 59.90 | 48.60 | 79.40 | 75.20 | 64.46 |
| | Arrow | 56.40 | 84.80 | 47.90 | 83.90 | 72.80 | 45.70 | 60.30 | 51.40 | 80.20 | 76.00 | 65.94 |
| | Arrow + MAPLE | 56.70 | 84.50 | 48.90 | 83.60 | 73.10 | 48.80 | 59.50 | 51.60 | 80.50 | 76.90 | **66.41** |
| | PHATGOOSE | 56.70 | 84.20 | 48.70 | 83.90 | 73.10 | 48.80 | 58.00 | 52.20 | 80.60 | 76.40 | 66.26 |
| | PHATGOOSE + MAPLE | 56.80 | 84.20 | 47.00 | 83.40 | 73.30 | 50.00 | 61.10 | 52.40 | 80.60 | 77.60 | **66.63** |
| LLAMA3 (8B) | Base | 51.80 | 73.30 | 48.20 | 68.40 | 78.10 | 36.60 | 52.90 | 41.60 | 80.40 | 73.00 | 60.40 |
| | Shared | 54.30 | 82.70 | 51.30 | 81.20 | 79.30 | 40.20 | 55.90 | 45.30 | 80.30 | 73.50 | 64.40 |
| | Uniform | 53.90 | 78.20 | 48.90 | 71.10 | 79.10 | 38.40 | 55.30 | 42.60 | 80.50 | 73.40 | 62.10 |
| | Arrow | 56.40 | 84.00 | 52.70 | 82.90 | 80.10 | 39.60 | 55.30 | 43.20 | 81.10 | 74.40 | 64.97 |
| | Arrow + MAPLE | 57.40 | 83.20 | 52.30 | 83.60 | 79.50 | 41.50 | 56.80 | 47.60 | 81.30 | 74.20 | **65.60** |
| | PHATGOOSE | 57.40 | 83.20 | 52.30 | 83.60 | 79.50 | 41.50 | 56.80 | 47.60 | 81.30 | 74.20 | 65.75 |
| | PHATGOOSE + MAPLE | 58.50 | 84.80 | 53.10 | 84.60 | 79.40 | 42.10 | 57.20 | 46.80 | 80.70 | 75.00 | **66.14** |

Table 1: Zero-shot accuracy (%) of **Phi-2 (2.8B)** and **LLaMA-3 (8B)** across 10 benchmarks under different routing strategies in comparison with MAPLE.

## 4.1 RESULTS

We empirically validate the effectiveness of MAPLE and compare it against various baselines with results summarized in table 1. As observed in table 1, applying MAPLE improves the prototype representations obtained by state-of-the-art routing methods (Arrow and PHATGOOSE), reflected by higher average zero-shot accuracy across various benchmarks. Because training updates only the LoRA prototypes, these gains arise from the improvement of the LoRA prototypes. The magnitude of improvement varies by model and routing scheme, likely reflecting different noise levels in prototypes learned by each method. The $\sim$0.5% gain from MAPLE's noisy-attribute elimination matches the $\sim$0.6% gap between strong baselines (Arrow vs. PHATGOOSE), suggesting that removing noisy attributes alone yields a meaningful improvement. Furthermore, improvements hold for both a small model (Phi-2) and a larger model (Llama-3), indicating robustness across parameter scales. Different benchmarks exhibit varying degrees of improvement, with the highest performance gain (4.4%) observed for OpenBookQA with arrow prototypes obtained on Llama3. This variability indicates that different OOD distributions are affected by noisy attributes to different degrees.

## 4.2 ANALYSIS AND ABLATIONS

In this section, we validate the design choices behind MAPLE and also re-examine the difference in routing entropy using refined prototypes for ID and OOD tasks, which was the primary indicator of inadequacy of LoRA prototypes.

### 4.2.1 EFFECT OF MASKING

Masking was introduced in MAPLE to enhance the elimination of noisy attributes embedded in the LoRA prototypes. To validate that this masking is necessary for the elimination of noisy attributes, we perform an ablation study wherein we disable masking(i.e., the target-task LoRA participates in routing and loss minimisation), which is compared against the case in which target-task LoRA is masked during training(MAPLE). We also report the results obtained via routing using the prototypes before any training for reference. The result of this ablation is summarised in table 2.

|  | Avg. Accuracy (%) |
|---|---|
| Arrow | 65.94 |
| MAPLE (Without masking) | 66.06 |
| MAPLE | **66.41** |

Table 2: Ablation on the effect of masking during MAPLE training. Masking yields a +0.41% improvement, validating our design choice and its significance in prototype refinement.

In the absence of masking, zero-shot accuracy shows only a marginal improvement of 0.12% with respect to initialization (Arrow), while enabling masking of the target task LoRA resulted in an improvement of 0.41%, validating the design choice of MAPLE as well as providing supportive evidence for our initial hypothesis of gradient suppression on inclusion of target-task LoRA.

### 4.2.2 IMPACT ON PROTOTYPE SIMILARITY

To assess how relationships between prototypes change under the proposed training scheme, we randomly sample nine LoRA prototypes and track how their cosine similarity evolves post-training for a randomly chosen layer. We summarise the pairwise cosine similarity between prototypes before and after training in figure 3. Comparing these similarities, we observe that after refining the prototypes with MAPLE, the pairwise cosine similarity between prototypes generally decreases by a small margin. We hypothesize that this results from removing shared noisy attributes encoded in LoRA prototypes, thereby reducing spurious dependencies between prototypes. The modest shift in cosine similarity suggests that noise encoded in LoRA prototypes is limited, which aligns with the average performance gain of 0.5% achieved by MAPLE.

### 4.2.3 ROUTING ENTROPY ANALYSIS

In section 3.1, we used routing entropy as a proxy for prototype quality and showed that routing entropy using LoRA prototypes obtained from popular methods (ARROW and PHATGOOSE) is similar for ID and OOD tasks.

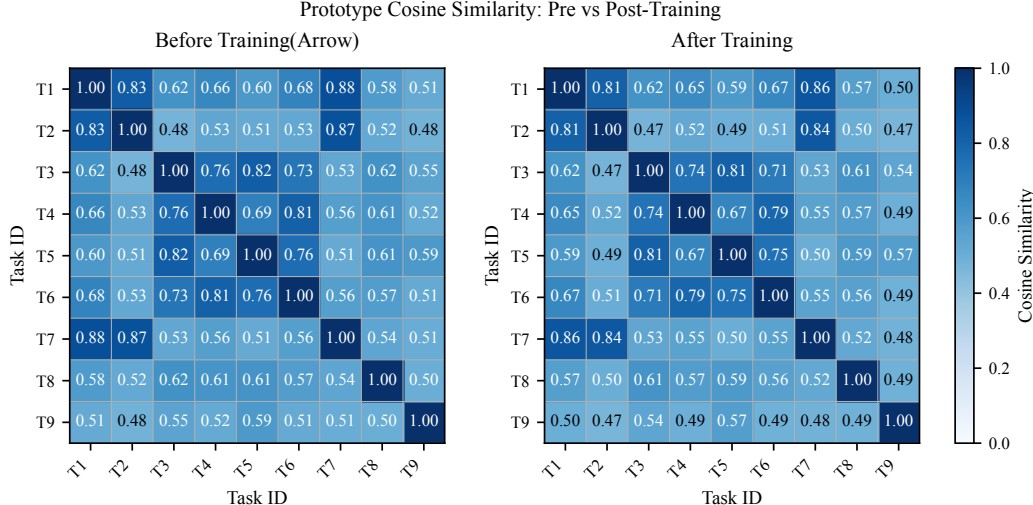

Figure 3: Pairwise cosine similarity of LoRA prototypes before and after training. Overall, the cosine similarity shows a decreasing trend post-training, indicating that the removal of spurious, noisy attributes makes the prototypes more orthogonal.

In this section, we revisit this analysis to assess how prototypes refined by MAPLE alter the routing entropy. Although MAPLE does not explicitly optimise routing entropy, improved representations should manifest as (i) lower absolute entropy and (ii) a larger ID–OOD separation. With MAPLE-trained prototypes, figure 4 shows a clear separation in routing entropy for ID and OOD inputs for both ARROW and PHATGOOSE initializations. ID entropy is consistently lower than OOD entropy, indicating effective identification of relevant LoRA as a result of improved discriminability of LoRA prototypes. Moreover, relative to the corresponding intilisations in figure 1, both ID and OOD entropies decrease, indicating more confident routing overall. Taken together, these results suggest that MAPLE yields more discriminative prototypes, validating its effectiveness in refining LoRA prototypes.

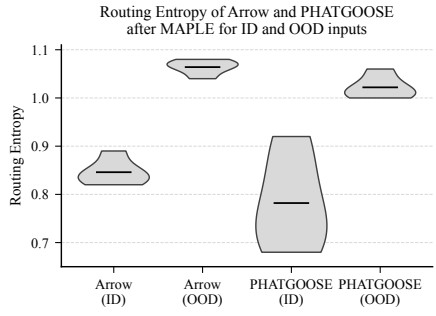

Figure 4: Routing entropy of learned representations with MAPLE using ARROW and PHATGOOSE initialisations across five tasks. ID and OOD distributions are clearly separated; ID routing entropy is consistently lower than OOD, indicating stronger discriminative capacity and more effective routing.

## 5 CONCLUSION AND FUTURE WORK

We demonstrate that LoRA prototypes learned in isolation fail to discriminate under joint routing: ID and OOD inputs exhibit similar routing entropy, and we formalize this with a theoretical analysis that exposes noisy subspaces. To address this, we introduce **MAPLE**, which masks the target adapter during prototype learning to refine LoRA prototypes by eliminating noisy attributes encoded in the LoRA representation. MAPLE improves LoRA prototypes, lowers routing entropy, and improves OOD routing across model scales and heterogeneous pools of pretrained LoRAs. The method is model-agnostic and refines representations produced by existing techniques. For future work, we will design objectives that penalise noisy subspaces, develop routing algorithms that are unaffected by noisy attributes encoded in prototypes, extend to multi-task and continual settings with evolving adapter pools, and scale to larger backbones and modalities (e.g., vision–language) to test generality and transfer.

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

## A    PROOFS AND DERIVATIONS

**Lemma 1.** *Let $\mathcal{B}$ be the matrix formed with the basis vectors $[w_1, w_2, ..., w_n]$ of a vector space $\mathcal{V}$ and $\mathbb{S}$ be the set of matrices corresponding to the bases of all subspaces of $\mathcal{V}$. Let $s^* \in \mathbb{S}$ be the basis matrix corresponding to the dominating subspace in $\mathcal{V}$ such that:*

$$\forall s \neq s^* \in \mathbb{S} \mid s \perp s^*; \|s^*\| >> \|s\|.$$

*Now, if $\exists p, q \in \mathbb{S} \mid |p \cdot q|, |p \cdot s^*|, |q \cdot s^*| \leq \epsilon$,*

*i.e., $p, q$, and $s^*$ are sufficiently orthogonal to each other, and $\mathtt{erank}(p + q) > \mathtt{erank}(s^*)$, then:*

$$\mathtt{erank}(\mathcal{B} - s^*) > \mathtt{erank}(\mathcal{B}),$$

*where $\mathtt{erank}(\cdot)$ is the effective rank of a matrix (Roy & Vetterli, 2007) and $\epsilon$ is a constant.*

*Proof.* According to Roy & Vetterli (2007), the effective rank of the matrix $B$ is defined as the exponentiated entropy of the distribution of its normalised singular values, *i.e.*,

$$\mathtt{erank}(B) = e^{H(\rho)},$$

$$\rho = [\rho_1, \rho_2, ..., \rho_n] = \left[ \frac{\sigma_1}{\sum_j \sigma_j}, \frac{\sigma_2}{\sum_j \sigma_j}, ..., \frac{\sigma_n}{\sum_j \sigma_j} \right]$$

where $\sigma = \{\sigma_1, \sigma_2, ..., \sigma_n\}$ are the singluar values of $\mathcal{B}$. It can be observed that the effective rank is proportional to the entropy of $\rho$, *i.e.*, $H(\rho)$. Let the $k^{\text{th}}$ singular value $\sigma_k = \sigma_{\mathcal{B}}^*$ be the largest singular value of $\mathcal{B}$ corresponding to the dominating subspace $s^*$. Then, the probability masses in $\rho$ can be expressed as:

$$\rho = [\rho_1, \rho_2, ..., \rho_n] = \left[ \frac{\sigma_1}{\sum_{j \neq k} \sigma_j + \sigma_{\mathcal{B}}^*}, \frac{\sigma_2}{\sum_{j \neq k} \sigma_j + \sigma_{\mathcal{B}}^*}, ..., \frac{\sigma_n}{\sum_{j \neq k} \sigma_j + \sigma_{\mathcal{B}}^*} \right]$$

Since we know that $\forall s \neq s^* \in \mathbb{S} \mid s \perp s^*; \|s^*\| >> \|s\|$ and $\exists p, q \in \mathbb{S} \mid |p \cdot q|, |p \cdot s^*|, |q \cdot s^*| \leq \epsilon$, without loss of generality, it is reasonable to assume that:

$$\forall i \neq k, \rho_i << 1/n; \rho_k >> 1/n \tag{2}$$

Further, due to the same given condition, upon the removal of $s^*$ from $\mathcal{B}$, the new set of singular values would be:

$$\sigma' = \{\sigma_1 + \epsilon_1, ..., \sigma_{k-1} + \epsilon_{k-1}, \sigma_{k+1} + \epsilon_{k+1}, ..., \sigma_n + \epsilon_n\} \cup \{\sigma_{\mathcal{B}}^* - \epsilon^*\}$$

where $\epsilon^* = \epsilon_1 + ... + \epsilon_{k-1} + \epsilon_{k+1} + ... + \epsilon_n$ is the decrease in the magnitude of $\sigma_{\mathcal{B}}^*$ due to the removal of $s^*$. Let $\rho' = [\rho_1', \rho_2', ..., \rho_n']$ be the singular values of $\mathcal{B} - s^*$. Therefore, due to Eq. (2), we have for all $i \neq k$:

$$-\rho_i' \ln \rho_i' = -\frac{\sigma_i + \epsilon_i}{\sum_{j \neq k} \sigma_j + \sigma_{\mathcal{B}}^*} \ln \frac{\sigma_i + \epsilon_i}{\sum_{j \neq k} \sigma_j + \sigma_{\mathcal{B}}^*} > -\frac{\sigma_i}{\sum_{j \neq k} \sigma_j + \sigma_{\mathcal{B}}^*} \ln \frac{\sigma_i}{\sum_{j \neq k} \sigma_j + \sigma_{\mathcal{B}}^*}$$

$$\implies -\rho_i' \ln \rho_i' > -\rho_i \ln \rho_i$$

$$-\rho_k' \ln \rho_k' = -\frac{\sigma_k - \epsilon}{\sum_{j \neq k} \sigma_j + \sigma_{\mathcal{B}}^*} \ln \frac{\sigma_k - \epsilon}{\sum_{j \neq k} \sigma_j + \sigma_{\mathcal{B}}^*} > -\frac{\sigma_k}{\sum_{j \neq k} \sigma_j + \sigma_{\mathcal{B}}^*} \ln \frac{\sigma_k}{\sum_{j \neq k} \sigma_j + \sigma_{\mathcal{B}}^*}$$

$$\implies -\rho_k' \ln \rho_k' > -\rho_k \ln \rho_k$$

$$\implies -\sum_{i \neq k} \rho_i' \ln \rho_i' - \rho_k' \ln \rho_k' > -\sum_{i \neq k} \rho_i \ln \rho_i - \rho_k \ln \rho_k$$

$$\implies H(\rho') > H(\rho) \implies e^{H(\rho')} > e^{H(\rho)}$$

$$\implies \mathtt{erank}(\mathcal{B} - s^*) > \mathtt{erank}(\mathcal{B})$$

This completes the proof of the lemma. □

The above lemma states the scenario when the presence of dominating dimensions in a basis could reduce the effective rank. This is subject to the condition that the more orthogonal the lower-dimensional subspaces, the greater is the reduction caused by the presence of the dominating dimensions. In other words, it predicts an increase in the effective rank upon the removal of dominating dimensions. The proof is based on the fact that when there exists a dominating dimension, the entropy of the SVD of the basis would be lower, and upon its removal, the entropy should increase.

**Lemma 2.** *The effective rank* $\texttt{erank}(\mathcal{B})$ *of an encoder with basis* $\mathcal{B}$ *is inversely proportional to the ID routing entropy,* $H(P_{\mathcal{B}}(X^i))$, *and directly proportional to the OOD routing entropy,* $H(P_{\mathcal{B}}(X^o))$.

*Proof.* According to Lemma 1, when $\eta$ is sufficiently orthogonal to all other subspaces of $\mathcal{B}$ and has a high enough norm, its removal would increase the effective rank. For a given effective rank $\texttt{erank}(\mathcal{B})$, we have for source distributions (ID):

$$\exists w^* \in \mathcal{B}, X^i \in \mathcal{D} \mid (w^* \cdot x - w \cdot x) > \gamma, \ \forall w \neq w^* \in \mathcal{B}, \forall x \in X_i$$

meaning that certain subspaces draw the majority of the mass towards themselves, leading to a low ID routing entropy, $H(P_{\mathcal{B}}(X^i))$. For target distributions (OOD), we have:

$$\forall w \in \mathcal{B}, X^o \in \mathcal{D} \mid w \cdot x < \epsilon,$$

as all features are roughly equally likely to match OOD features, implying a high OOD routing entropy, $H(P_{\mathcal{B}}(X^o))$. Now, let a subspace with basis $\eta = [\eta_1, \eta_2, ..., \eta_n]$ be removed from $\mathcal{B}$ leading to an increase in the effective rank as in Lemma 1, *i.e.*, $\texttt{erank}(\mathcal{B} - \eta) > \texttt{erank}(\mathcal{B})$. Then, the ID routing entropy would become:

$$\exists (w_i - \eta_i)^* \in (\mathcal{B} - \eta), X^i \in \mathcal{D} \mid ((w_i - \eta_i)^* \cdot x - (w_j - \eta_j) \cdot x) > \gamma' > \gamma,$$
$$\forall (w_i - \eta_i) \neq (w_j - \eta_j)^* \in \mathcal{B} - \eta, \forall x \in X_i,$$

implying a decreased ID routing entropy. For target distributions routed through $\mathcal{B} - \eta$, we have:

$$\forall (w_i - \eta_i) \in \mathcal{B}, X^o \in \mathcal{D} \mid (w_i - \eta_i) \cdot x < \epsilon' < \epsilon,$$

implying an increased OOD routing entropy as the probability masses get more spread out across the different $(w - \eta)$s. This completes the proof of the lemma. $\qquad\square$

### A.1 PROOF OF THEOREM 1

*Proof.* Let $B_1 = \mathcal{B} - \eta_1$ and $B_2 = \mathcal{B} - \eta_2$. We start with the premise that $\texttt{erank}(B_1) > \texttt{erank}(B_2)$. Therefore:

$$\frac{1}{\texttt{erank}(B_1)} < \frac{1}{\texttt{erank}(B_2)} \implies -\frac{1}{\texttt{erank}(B_1)} > -\frac{1}{\texttt{erank}(B_2)} \tag{3}$$

$$\implies \texttt{erank}(B_1) - \frac{1}{\texttt{erank}(B_1)} > \texttt{erank}(B_2) - \frac{1}{\texttt{erank}(B_2)} \tag{4}$$

Now, we know from Lemma 2 that for any basis $B$, $\texttt{erank}(B) \propto H(P_B(X^o))$ and $1/\texttt{erank}(B) \propto H(P_B(X^i))$. Substituting this in Eq. (4):

$$H(P_{B_1}(X^o)) - H(P_{B_1}(X^i)) > H(P_{B_2}(X^o)) - H(P_{B_2}(X^i)) \tag{5}$$

Dividing Eq. (5) by $H(P_{\mathcal{B}}(X^o)) - H(P_{\mathcal{B}}(X^i))$:

$$\frac{H(P_{B_1}(X^o)) - H(P_{B_1}(X^i))}{H(P_{\mathcal{B}}(X^o)) - H(P_{\mathcal{B}}(X^i))} > \frac{H(P_{B_2}(X^o)) - H(P_{B_2}(X^i))}{H(P_{\mathcal{B}}(X^o)) - H(P_{\mathcal{B}}(X^i))}$$
$$\implies r(\eta_1) > r(\eta_2)$$

This completes the proof of the theorem. $\qquad\square$

## A.2 PROOF OF THEOREM 2

*Proof.* Let $D_1, D_2, ..., D_m$ be an arbitrary discretization of the distribution space $\mathcal{D}$. Let the aggregate similarities of $\eta$ with each of these subspaces be given by:

$$\mathbb{S} = \{D_1 \odot \eta, D_2 \odot \eta, ..., D_m \odot \eta\}$$
$$= \{s_1, s_2, ..., s_m\},$$
$$s_i = D_i \odot \eta = \frac{1}{|D_i|} \int_{x \in D_i} x \cdot \eta \, dx$$

Without loss of generality, consider a partitioning of $\mathbb{S}$ into the following disjoint sets:

$$S = \{s_1, s_2, ..., s_k\} \mid \forall s \in S, s > \gamma$$
$$\bar{S} = \{s_k, s_{k+1}, ..., s_m\} \mid \forall s \in S, s < \epsilon$$

In other words, $S$ is the subset of $\mathcal{D}$ for which $\eta$ is relevant, and $\bar{S}$ is the completementary subset of $\mathcal{D}$ for which $\eta$ is noisy. Therefore, the noisiness of $\eta$ *wrt* $\mathcal{D}$ can be given by:

$$\mathcal{N}(\eta) = \frac{|\bar{S}|}{|S + \bar{S}|}$$

Now, we know the following:

$$\frac{|\bar{S}|}{|S|} \propto \frac{1}{\delta}, \quad \frac{\gamma}{\epsilon} \propto \frac{1}{\delta},$$

Combining the above, we have:

$$\frac{\gamma}{\epsilon} \frac{|\bar{S}|}{|S|} \propto \frac{1}{\delta} \implies \frac{|\bar{S}|}{|S|} \propto \frac{\epsilon}{\gamma} \frac{1}{\delta} \tag{6}$$

$$\implies \frac{|\bar{S}|}{|S|} + 1 \propto \frac{\epsilon}{\gamma\delta} + 1 \implies \frac{|S + \bar{S}|}{|S|} \propto \frac{\epsilon + \gamma\delta}{\gamma\delta} \tag{7}$$

Dividing Eq. (6) by Eq. (7):

$$\frac{|\bar{S}|}{|S + \bar{S}|} \propto \frac{\epsilon}{\epsilon + \gamma\delta} \implies \mathcal{N}(\eta) \propto \delta^{-1}$$

This completes the proof of the theorem. □

## B REPRODUCIBILITY

In this section, we report all model- and dataset-specific hyperparameters in Table 3. Experiments were executed on two NVIDIA A100 GPUs. To enable exact replication, we release a clean, user-oriented codebase and pretrained checkpoints. All datasets used in this work are publicly available. For all methods, we use the same top-k as the original routing scheme (k=2 for PHATGOOSE and k=4 for arrow) to ensure the same inference compute capacity. In zero-shot downstream evaluation, we append an EOS token to the target options to indicate the end of a sentence. For multiple-choice task evaluations, we select continuations using scores normalized by token length. For all LoRA experts trained in this work, we use a LoRA rank of 4, a dropout probability of 0.05, an $\alpha$ value of 16, and a learning rate of 1e-4, along with learning rate warm-up and annealing phases, the same setting as in Ostapenko et al. (2024). For all the models, we modified attention layers and attention output projection, with weight sharing among LoRAs applied to attention layers.

## C LIMITATIONS

MAPLE is effective in eliminating noisy attributes encoded in LoRA prototypes, but the sole improvement arises from the elimination of noisy components involved in routing. The amount of

Table 3: Hyperparameters for the Phi-2 and Llama 3 models. Replace placeholder values as appropriate for your setup.

| Parameter | Phi-2 | Llama 3 |
|---|---|---|
| Learning rate (LR) | 1e-4 | 1e-3 |
| Batch size | 16 | 16 |
| Epochs | 1 | 1 |
| warmup_proportion | 0.06 | 0.06 |

noise encoded in LoRA prototypes depends on various factors, such as the Quality of data used for LoRA training, the similarity of the tasks used for training LoRAs, as well as the technique used to obtain LoRA representations. In case the LoRA prototypes don't encode many noisy attributes, the performance improvement with MAPLE would be very marginal. Since MAPLE is a training-based method, it requires data(only a small fraction), which demands access to the LoRA training data, which may not always be accessible, restricting its utility.

## D  LLM USAGE

LLMs were utilized in refining the paper, primarily for rephrasing, restructuring, and condensing sentences. They were also queried for suggestions on additional experiments; however, none of these suggestions were deemed suitable for inclusion, limiting the use of LLMs to writing assistance only.

## E  ADDITIONAL RESULTS

### E.1  MAPLE TRAINING DYNAMICS

To provide further insights into the training dynamics of MAPLE, we provide the evolution of validation and testing loss of MAPLE across training iterations in Fig. 5. As shown in Fig. 5, both losses decrease relative to the Arrow initialization at iteration 0, exhibiting clear and structured trajectories rather than stochastic fluctuations. These trends indicate that MAPLE's improvements arise from meaningful optimization behavior and are not artifacts of random variation.

### E.2  EFFECT OF MASKING IN DENOISING LoRA PROTOTYPES

To illustrate the role of masking, we compare two settings: MAPLE and a baseline without masking. Consider $k$ pretrained LoRAs, each trained on a corresponding task and therefore encoding both useful task structure and task-specific noise.

Without masking, an input from task $i$ will most likely route to LoRA $i$. Because LoRA $i$ closely matches the data distribution of task $i$, the model attains low loss and therefore receives weak gradients. This limits its ability to remove task-specific noise embedded within LoRA $i$. Moreover, because the noise in LoRA $i$ aligns with noise in the task-$i$ inputs, the routing mechanism does not incur penalties that would otherwise encourage denoising.

With masking applied, LoRA $i$ is excluded when processing inputs from task $i$, forcing the model to rely on LoRAs trained on other tasks. The noise patterns in these LoRAs do not match the noise present in task $i$, leading to routing errors and higher loss. These errors produce stronger gradients, which push the shared representation space toward reducing noise. Consequently, masking enables MAPLE to learn more task-agnostic and better-generalizing prototypes.

Empirical evidence for this mechanism is shown in Fig. 6. During training, MAPLE exhibits higher ID loss (Fig. 6 (Left)) under masking than in the no-masking baseline. On OOD data(Fig. 6 (Right)), MAPLE achieves consistently lower loss, demonstrating more effective removal of task-specific artifacts. This behavior aligns with the improvements reported in Table 2 of the main text, where masking outperforms the no-masking variant.

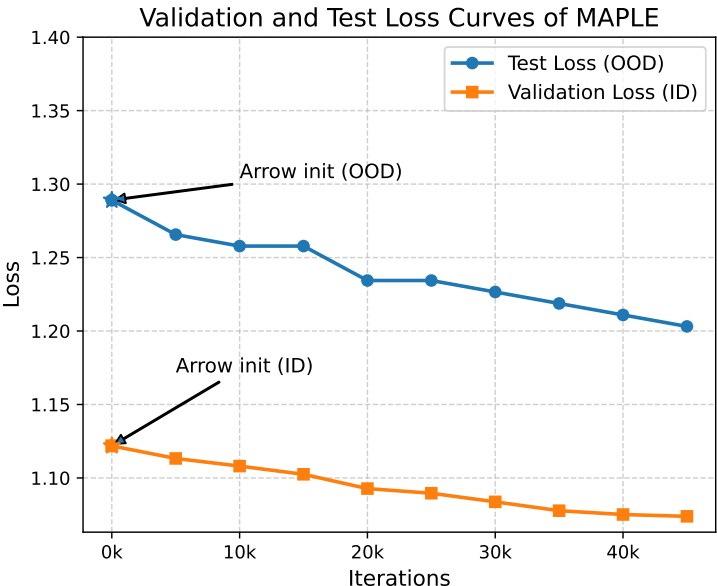

Figure 5: Validation (ID) and test (OOD) loss curves for MAPLE training. Iteration 0 corresponds to the Arrow initialization, highlighted by star markers for both ID and OOD. MAPLE fine-tuning steadily reduces both validation and test loss relative to the Arrow initialization.

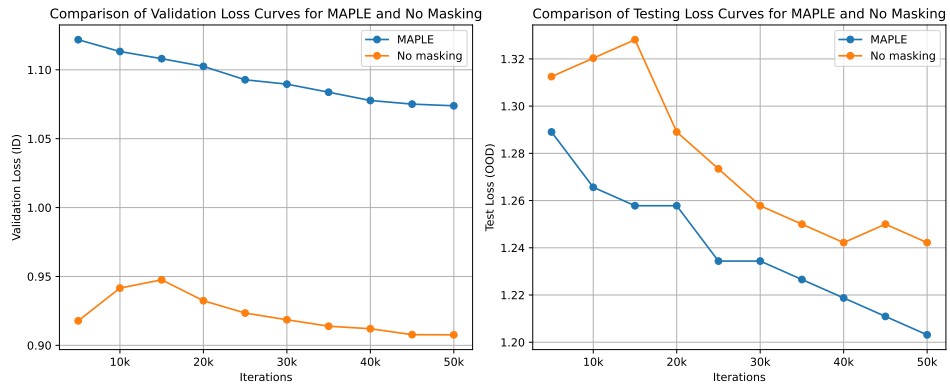

Figure 6: Validation and test loss curves comparing the MAPLE masking strategy with a no-masking baseline. The left panel shows in-distribution validation behavior across training iterations, while the right panel presents out-of-distribution test performance. The higher loss on ID samples of MAPLE

## F  GENERALIZATION

**Definition 2** (Membership Coefficient). The membership coefficient of a distribution $X$ with respect to a basis $\mathcal{B}$ is the smallest real number $m$ such that:

$$\max_{w_i \in \mathcal{B}} \int_{x \in X} \frac{w_i \cdot x}{\sum_{j=1}^{n} w_j \cdot x} \, p(x) \, dx \le m$$

where $p(\cdot)$ is the probability density function over $X$. In other words, $m$ is a measure of how similar $X$ is to any of the basis vectors in $\mathcal{B}$. The membership coefficient directly corresponds to distributional membership – the lower the value of $m$, the more out-of-distribution $X$ is, and vice versa.

In the subsequent results, we use the following shorthand to denote the mass on the $i$-th basis vector in a routing plan of a distribution $X$ over a basis $\mathcal{B}$:

$$P_{\mathcal{B}}(X)\{w_i\} = \int_{x \in X} \frac{w_i \cdot x}{\sum_{j=1}^{n} w_j \cdot x} \, p(x) \, dx = w_i \odot X$$

**Setting:** Let $\mathcal{B}$ be the basis learnable from $X^i$ (ID training samples) that is optimal in terms of ID and OOD generalization. All membership co-efficients referred to henceforth are with respect to this optimal basis $\mathcal{B}$. Let $S = (X_1, X_2, ..., X_n)$ be a sequence of distributions with corresponding membership coefficients $(m_1, m_2, ..., m_n)$ in the decreasing order over the basis $\mathcal{B}$ such that the following conditions hold:

1. $X_1$ has the highest possible and $X_n$ has that lowest possible values of the membership coefficient in $\mathcal{B}$.

2. For any pair of distributions $X_a, X_b, a < b$ and any basis $\hat{\mathcal{B}} \subset \mathcal{B}$ such that $P_{\mathcal{B}}(X_a) = P_{\hat{\mathcal{B}}}(X_a)$, then, for any $w \in \operatorname{support}(P_{\hat{\mathcal{B}}}(X_a))$:

$$w \cdot X_a = w \cdot X_b$$

3. For any pair of distributions $X_a, X_b, a < b$ and any basis $\hat{\mathcal{B}} \subset \mathcal{B}$ such that $P_{\mathcal{B}}(X_b) = P_{\hat{\mathcal{B}}}(X_b)$, then, for any $w \in \operatorname{support}(P_{\hat{\mathcal{B}}}(X_b)) \setminus \operatorname{support}(P_{\hat{\mathcal{B}}}(X_a))$:

$$w \cdot X_a = 0$$

4. Admits a monotone-support ordering: If $a < b$, then $\operatorname{support}(P_{\mathcal{B}}(X_a)) \subset \operatorname{support}(P_{\mathcal{B}}(X_b))$.

**Lemma 3.** *If a distribution $X_n$ has a membership coefficient $m_n$ in $\mathcal{B}$, then for any basis $\hat{\mathcal{B}} \subset \mathcal{B}$, the optimal routing plan for $X_n$ in $\hat{\mathcal{B}}$ is given by:*

$$P_{\hat{\mathcal{B}}}(X_n) = \{w_i \odot X_n\}_{i=1}^{k},$$

*where*

$$k = \begin{cases} \dim(\hat{\mathcal{B}}) & \text{if } |\operatorname{support}(P_{\mathcal{B}}(X_n))| > \dim(\hat{\mathcal{B}}) \\ |\operatorname{support}(P_{\mathcal{B}}(X_n))| & \text{otherwise,} \end{cases}$$

*assuming without loss of generality (and only to keep the proofs simple without affecting the general results) that $w_i$ corresponds to the same basis vector in $\mathcal{B}$ and $\hat{\mathcal{B}}$.*

*Proof.* Since $\mathcal{B}$ is a complete basis, the result when $|\operatorname{support}(P_{\mathcal{B}}(X_n))| \leq \dim(\hat{\mathcal{B}})$ is trivial, since the routing plan of $X_n$ would be exactly the same in both $\mathcal{B}$ and $\hat{\mathcal{B}}$.

When $|\operatorname{support}(P_{\mathcal{B}}(X_n))| > \dim(\hat{\mathcal{B}})$, since $\hat{\mathcal{B}} \subset \mathcal{B}$, closest possible alignment of routing distributions, *i.e.*, $\min \operatorname{KL}(P_{\hat{\mathcal{B}}}(X_n) || P_{\mathcal{B}}(X_n))$, can be achieved only when the input is routed through exactly the same basis vectors from $\mathcal{B}$ that are retained in $\hat{\mathcal{B}}$. Any other routing plan would lead to a higher $\operatorname{KL}(P_{\hat{\mathcal{B}}}(X_n) || P_{\mathcal{B}}(X_n))$ due to misalignment of the probability masses. In the lemma, without loss of generality, since the basis vectors are assumed to correspond between $\mathcal{B}$ and $\hat{\mathcal{B}}$, this property is automatically ensured in $P_{\hat{\mathcal{B}}}(X_n)$.

This completes the proof of the lemma. $\square$

**Definition 3** (Representibility). *A distribution $X$ is representable in a basis $\hat{\mathcal{B}} \subset \mathcal{B}$ iff:*

$$\operatorname{support}(P_{\mathcal{B}}(X)) \subseteq \hat{\mathcal{B}}$$

**Lemma 4.** *Let $X_{min}$ be the distribution having the highest entropy routing plan $P_{\hat{\mathcal{B}}}(X_{min})$ with respect to a basis $\hat{\mathcal{B}} \subset \mathcal{B}$ and some membership coeffcient $m_{min}$ such that $\hat{\mathcal{B}} = \operatorname{support}(P_{\hat{\mathcal{B}}}(X_{min}))$. In other words, $\hat{\mathcal{B}}$ is the smallest subset of $\mathcal{B}$ (the entirety of) which is needed to represent $P_{\mathcal{B}}(X_{min})$. If there exists a distribution $X_{out}$ such that it has a lower membership coefficient than $m_{min}$, i.e., $m_{min} > m_{out}$ but has the same routing entropy as $X_{min}$, i.e., $H(P_{\hat{\mathcal{B}}}(X_{min})) = H(P_{\hat{\mathcal{B}}}(X_{out})) > 0$, then $X_{out}$ is not representable in $\hat{\mathcal{B}}$, where $X_{out}$ is representable in $\hat{\mathcal{B}}$ if $P_{\mathcal{B}}(X_{out}) = P_{\hat{\mathcal{B}}}(X_{out})$.*

*Proof.* Since $X_{\text{out}}$ has a membership coefficient of $m_{\text{out}}$, and $X_{\text{min}}$ has a membership co-efficient of $m_{\text{min}}$, their respective routing plan in $\hat{\mathcal{B}}$ are given by:

$$P_{\hat{\mathcal{B}}}(X_{\text{out}}) = \{w_1 \odot X_{\text{out}}, w_2 \odot X_{\text{out}}, ..., w_j \odot X_{\text{out}}\}$$
$$P_{\hat{\mathcal{B}}}(X_{\text{min}}) = \{w_1 \odot X_{\text{min}}, w_2 \odot X_{\text{min}}, ..., w_i \odot X_{\text{min}}\}$$

Since we know that $m_{\text{min}} > m_{\text{out}}$, we have:

$$P_{\hat{\mathcal{B}}}(X_{\text{out}}) = \{w_1 \odot X_{\text{out}}, ..., w_i \odot X_{\text{out}}, w_{i+1} \odot X_{\text{out}}, ..., w_j \odot X_{\text{out}}\}$$

Now, since we know that:

- $H(P_{\hat{\mathcal{B}}}(X_{\text{min}})) = H(P_{\hat{\mathcal{B}}}(X_{\text{out}}))$,

- conditions 2, 3, and 4 described in the setting,

- $w_1 \odot X_{\text{out}}, w_2 \odot X_{\text{out}}, ..., w_i \odot X_{\text{out}} \neq 0$ as $H(P_{\hat{\mathcal{B}}}(X_{\text{min}})) > 0$,

- $\text{span}(\hat{\mathcal{B}}) = \text{support}(P_{\hat{\mathcal{B}}}(X_{\text{min}}))$,

it must be the case that the vectors $w_{i+1}, w_{i+2}, ...w_j$ do not lie in $\hat{\mathcal{B}}$, *i.e.,*:

$$w_{i+1} \odot X_{\text{out}} = w_{i+2} \odot X_{\text{out}} = ... = w_j \odot X_{\text{out}} = 0,$$

implying that the corresponding components of $X_{\text{out}}$ cannot be represented in $\hat{\mathcal{B}}$.

This completes the proof of the lemma. $\square$

**Lemma 5.** *If a basis $\hat{\mathcal{B}} \in \mathcal{B}$ can represent a distribution $X_b \in S$, then it can represent all other distributions $X_a \in S$ such that $a < b$.*

*Proof.* Since $X_b$ can be represented in $\mathcal{B}$, its routing plan is given by:

$$P_{\hat{\mathcal{B}}}(X_b) = \{w_1 \odot X_b, w_2 \odot X_b, ..., w_j \odot X_b\}$$

where $w_1 \odot X_b, w_2 \odot X_b, ..., w_j \odot X_b \neq 0$. Now consider a distribution $X_a, a < b$. If routed through the same set of bases that $X_b$ was routed through, the plan would be:

$$\{w_1 \odot X_a, w_2 \odot X_a, ..., w_j \odot X_a\}$$
$$= \{w_1 \odot X_a, ..., w_i \odot X_a, w_{i+1} \odot X_a, ..., w_j \odot X_a\}$$

However, since $a < b$, the membership coefficient $m_a > m_b$. Therefore, since $\hat{\mathcal{B}} \subset \mathcal{B}$, according to the condition 3 in the setting and Lemma 3, it follows that:

$$w_{i+1} \odot X_a, ..., w_j \odot X_a = 0$$

Therefore, with the same set of bases vectors used to route $X_b$ in $\hat{\mathcal{B}}$, the routing plan for $X_a$ becomes:

$$\{w_1 \odot X_a, w_2 \odot X_a, ..., w_i \odot X_a\} = P_{\hat{\mathcal{B}}}(X_a),$$

the optimal routing plan for $X_a$ in $\hat{\mathcal{B}}$ through which $X_a$ is representable in $\hat{\mathcal{B}}$.

This completes the proof of the lemma. $\square$

**Theorem 3** (Generalization Spectrum). *Let $S = (X_1, X_2, ..., X_n)$ be a sequence of distributions with corresponding membership coefficients $(m_1, m_2, ..., m_n)$ in the decreasing order over the basis $\mathcal{B}$ such that $X_1$ has the highest possible and $X_n$ has that lowest possible values of the membership coefficient in $\mathcal{B}$ respectively. Let $\mathcal{B}_1 \subset \mathcal{B}_2 \subseteq \mathcal{B}$ be bases such that:*

$$H(P_{\mathcal{B}_1}(X_n)) - H(P_{\mathcal{B}_1}(X_1)) < H(P_{\mathcal{B}_2}(X_n)) - H(P_{\mathcal{B}_2}(X_1))$$

*Then, $\mathcal{B}_2$ generalizes over a wider spectrum of distributions in S than $\mathcal{B}_1$; or simply, $\mathcal{B}_2$ has better generalization than $\mathcal{B}_1$ over S.*

*Proof.* Since $X_1$ is completely in-distribution (highest possible value of the membership coefficient), according to the monotone-support ordering condition in the setting (condition 4), its support is contained in both $\mathcal{B}_1$ and $\mathcal{B}_2$. Additionally, due to conditions 2 and 3 in the setting, we have that $H(P_{\mathcal{B}_1}(X_1)) = H(P_{\mathcal{B}_2}(X_1))$. Then,

$$H(P_{\mathcal{B}_1}(X_n)) < H(P_{\mathcal{B}_2}(X_n)) \tag{8}$$

Now, the strict inequality in Eq. (8) means that for $\mathcal{B}_1$, there exists a distribution $X_i$ representable in $\mathcal{B}_1$ having a membership coefficient $m_i > m_n$ and $\text{support}(P_{\mathcal{B}_1}(X_i)) = \text{span}(\mathcal{B}_1)$. Now due to the condition 2 in the setting, we have:

$$H(P_{\mathcal{B}_1}(X_i)) = H(P_{\mathcal{B}_1}(X_n)), \tag{9}$$

implying based on Lemma 4 that all distributions $X_j, j > i$ are not representable in $\mathcal{B}_1$. Similarly, there must exist some $X_j$ representable in $\mathcal{B}_2$ with membership coefficient $m_i > m_j \geq m_n$ such that:

$$H(P_{\mathcal{B}_2}(X_j)) = H(P_{\mathcal{B}_2}(X_n)) \tag{10}$$

Combining Eqs. (8) to (10), we get:

$$H(P_{\mathcal{B}_1}(X_i)) < H(P_{\mathcal{B}_2}(X_j)) \tag{11}$$

However, since $X_i$ is the distribution that has the highest entropy in $\mathcal{B}_1$, following the same logic as that of Eq. (9):

$$H(P_{\mathcal{B}_1}(X_i)) = H(P_{\mathcal{B}_1}(X_j)), \tag{12}$$

implying again based on Lemma 4 that $X_j$ is not representable in $\mathcal{B}_1$. Therefore, based on Eqs. (11) and (12), there exists a distribution $X_j \in S$ with a low enough membership coefficient that is not representable in $B_1$ but is representable in $B_2$. In other words, due to the following inequality:

$$H(P_{\mathcal{B}_1}(X_n)) - H(P_{\mathcal{B}_1}(X_1)) < H(P_{\mathcal{B}_2}(X_n)) - H(P_{\mathcal{B}_2}(X_1))$$

$\mathcal{B}_1$ can only represent distributions $(X_1, ..., X_i)$, whereas $\mathcal{B}_2$ can additionally represent distributions $(X_{i+1}, ..., X_j)$ which cannot be represented in $\mathcal{B}_1$. Hence, based on Lemma 5, a greater number or a wider spectrum of distributions in $S$ are representable in $\mathcal{B}_2$, which consequently implies that it has better generalization than $\mathcal{B}_1$.

This completes the proof of the theorem. $\qquad\square$

**Intuition:** The intuition behind Theorem 3 is that a LoRA prototype basis that exhibits a relatively low ID entropy and high OOD entropy, *i.e.*, a larger gap between ID and OOD entropies, can represent a larger spectrum of distributions. This increases its chances of being complete (as per Definition 1) for this expanded spectrum of distributions. This ultimately implies better generalization since the label of an input can only be correctly predicted if it can be completely represented.