# OpenReview forum: "MAPLE: Masked Adapter Prototype Learning for OOD generalization"
_ICLR.cc/2026/Conference — Submitted to ICLR 2026_

### Official Review · Reviewer_8KQz · 2025-10-30

**Soundness:** 2
**Presentation:** 2
**Contribution:** 2
**Rating:** 4
**Confidence:** 3

**Summary:**

This paper aims to improve OOD methods that use multiple pre-trained LoRAs with routing. The authors utilize routing entropy to demonstrate that routing may not provide complementary information across different tasks. Furthermore, they present a theoretical analysis to explain the noisiness in routing and the resulting degradation in OOD performance. To address this issue, the authors propose removing the target-task LoRA from the forward pass during training, positing that this will help reduce spurious, task-specific noise in the learned representations.

**Strengths:**

This paper addresses an interesting and important problem. I like the comprehensive overview of prior works and their connection to the current study. The results also demonstrate good improvements over previous state-of-the-art methods.

**Weaknesses:**

From the problem formulation, it is not entirely clear what type of Out-of-Distribution (OOD) setting is used by the authors. Some parts of the paper would benefit from clearer sentences and a more precise problem formulation (see Questions for details). Additionally, certain ideas in the paper could be explained more clearly (see Questions for details).

The proposed method is based on removing the target-task LoRA from the forward pass during training. The authors posit that this will reduce noisiness; however, this assumption is not supported by prior work or analysis.

**Questions:**

It would be helpful to begin the methodology section with a brief introduction to the problem and the notations used. For instance, the authors introduce Equation 1 without defining $e$ and $x$, which makes it difficult to understand the purpose of routing entropy. Similarly, the section would benefit from first introducing the problem setting (e.g., multitask learning with LoRA and routing) and clearly explaining what prototypes and routing entropy are before using them in the analysis.

On line 164, authors mention following:

> We start by formalizing our argument that for a set of experts to generalize OOD, it must be accompanied by an increased entropy, relative to ID samples, when applied to OOD samples.

Entropy represents the expected information over a distribution. For a set of experts to generalize to OOD data, the entropy for in-distribution (ID) and out-of-distribution (OOD) samples should ideally be similar. In other words, the expected information for OOD samples should be comparable to that of ID samples. If OOD samples are highly unexpected (i.e., yield much higher information), the model may struggle to generalize. Perhaps the authors are instead referring to routing entropy? Please correct me if this interpretation is inaccurate.

In the line 143 and line 160, authors reason about routing entropy for IID and OOD samples.

> Ideally, for in-distribution (ID) tasks (i.e., LoRA trained on the input task exists in the LoRA pool), the routing entropy should be lower. While for out-ofdistribution (OOD) tasks, the routing entropy is expected to be higher, reflecting greater uncertainty.

And then explain observed similarity in entropy across ID and OOD samples.

> This could be due to representations encoding certain noisy attributes, which blur inter-adapter distinctions and thus yield high routing uncertainty.

However, similarity in routing entropy could also result from an effective router that performs well on OOD tasks. It would be helpful if the authors provided an intuitive explanation of this aspect.

On line 294, authors mention target-task.

> We hypothesize that this failure stems from including the target-task LoRA in the routing during training.

However, it is not quite clear what target-task in this setting means.

---

> ### Author Response · Authors · 2025-11-21
> **Rebuttal 1/2**
>
> We thank the reviewer for their valuable comments and suggestions, which will strengthen our paper. We address your questions and concerns below
>
> -----
>
> **W1/Q1:** From the problem formulation, it is not entirely clear what type of Out-of-Distribution (OOD) setting is used by the authors. It would be helpful to begin the methodology section with a brief introduction to the problem and the notations used. For instance, the authors introduce Equation 1 without defining $e$ and $x$, which makes it difficult to understand the purpose of routing entropy. Similarly, the section would benefit from first introducing the problem setting (e.g., multitask learning with LoRA and routing) and clearly explaining what prototypes and routing entropy are before using them in the analysis.
>
> **Response:** We thank the reviewer for identifying these ambiguities in our manuscript. We clarify them below.
>
> Given a set of pre-trained LoRAs, the goal is to enhance the OOD generalization capabilities of the base LLM. Our OOD setup is defined with respect to the pretrained LoRA pool. Samples from the tasks used to train these LoRAs constitute the in-distribution (ID) samples, while samples from any task outside this set are treated as OOD.
>
> **Problem Setting:** We consider an LLM equipped with a collection of pretrained LoRA adapters. The goal is to exploit this pool to achieve improved out-of-distribution (OOD) generalization. Specifically, given an input, the model must route among an appropriate subset of LoRAs from the pretrained pool such that their combined adaptation yields robust performance on OOD inputs. This formulation frames OOD generalization as a routing problem over a fixed library of specialized, lightweight LoRAs.
>
> We hypothesize noise encoded in LoRA representations as a limiting factor for OOD generalization, and aim to eliminate these noise components from the LoRA representations to improve OOD generalization.
>
> **Routing entropy:** measures the uncertainty in selecting an expert for a token. Lower routing entropy indicates more decisive routing, while higher routing entropy reflects greater uncertainty. In Eq. 1, $e$ denotes an expert/LoRA, $x$ represents the input token, and $M$ is the total number of pretrained experts/LoRAs in the pool.
>
> We will revise the manuscript to clarify this distinction and reorganize the relevant sections to improve conceptual flow. Thank you for pointing out these issues and helping us to improve the paper.
>
> -----
>
> **W2:** The proposed method is based on removing the target-task LoRA from the forward pass during training. The authors posit that this will reduce noisiness; however, this assumption is not supported by prior work or analysis.
>
> **Response:** Thank you for identifying this gap in our work. To validate our assumption that the MAPLE training paradigm eliminates noisy components rather than useful signals, we perform the following analysis. We consider three sets of prototype vectors:
> - **Pre-finetuning prototypes(W⁽⁰⁾)** (e.g., Arrow),
> - **Post-finetuning prototypes(W⁽*⁾)** (MAPLE),
> - **Update (Δ) prototypes**, defined as the difference between pre- and post-finetuning prototypes: Δ = W⁽⁰⁾ − W⁽*⁾.
>
> For each of these sets, we (i) compute the dot product $(a \cdot b)$ between normalized prototypes and normalized input tokens, and (ii) measure the corresponding fraction of representation energy along each prototype direction, given by $(a \cdot b)^2$. The results, averaged over tokens, layers, and experts, are summarized below:
>
> | Prototype | $(a \cdot b)$ | Energy $(a \cdot b)^2$ |
> |-----------|---------------|---------------------------|
> | Arrow     | 0.273         | 7.45%                     |
> | MAPLE     | 0.272         | 7.40%                     |
> | Δ         | 0.136         | 1.85%                     |
>
> We observe that the average cosine similarity with Arrow and MAPLE prototypes is 0.273 and 0.272, respectively (≈7.4% of representation energy), while the average similarity with the eliminated directions Δ is only 0.136 (≈1.9% energy), i.e., about **4× lower**. This shows that MAPLE preserves the task-aligned prototype geometry of the initialization eg, Arrow (their alignment with the data remains essentially unchanged). At the same time, the directions it removes are low-energy, weakly activated directions. This is consistent with our interpretation of the eliminated subspace as being largely noisy or task-irrelevant, rather than containing useful task-specific information.
>
> -----
>
> **Q2:** Perhaps the authors are instead referring to routing entropy? Please correct me if this interpretation is inaccurate.
>
> **Response:** Yes, indeed, we were referring to routing entropy. The Reviewers' interpretation is correct, even though in our context we are always referring to routing entropy. We will revise the manuscript to make this explicit.
>
> -----

---

> ### Author Response · Authors · 2025-11-21
> **Rebuttal 2/2**
>
> **Q3:** However, similarity in routing entropy could also result from an effective router that performs well on OOD tasks. It would be helpful if the authors provided an intuitive explanation of this aspect.
>
> **Response:** It is indeed true that similarity in routing entropy could also result from the router performing equally well on both ID and OOD tasks. However, as observed in prior works following from the No Free Lunch theorem (Wolpert & Macready, 1997), it is impossible to design a learning algorithm that exhibits low error (and hence, similar routing entropy) across all distributions [a]. In other words, if a model has low error in one distribution, there must be another distribution where its error is high. Another interpretation of this idea is that when we consider the mean population risk across all distributions, it is impossible to do better than a learner that is optimal on average [b].
>
> Given this, we posit that similar routing entropy can arise only from misdirected target routing, since, from prior theoretical results, we know that it is impossible to have a model that performs better than average out of distribution, i.e., when we cannot make any distributional assumptions. We therefore aim to fine-tune the routing mechanism such that with increasing uncertainty about the input distribution (increasing OOD), the routing mechanism should rely less and less on any single expert (equivalent to the optimal on average model). This would consequently lead to an increase in routing entropy as samples become increasingly OOD.
>
> -----
>
> **Q4:** We hypothesize that this failure stems from including the target-task LoRA in the routing during training. It is not quite clear what the target task in this setting means.
>
> **Response:** During maple training, we sample data that were used to train the LoRAs and use these data to improve LoRA representations. Thus, for any data point in the training set, there exists a LoRA that was trained with this data point. The “target-task LoRA” refers to the LoRA that was trained on that specific datapoint.
>
> -----
>
> References:\
> [a] Gulrajani, "In Search of Lost Domain Generalization", ICLR 2021.\
> [b] Arjovsky et al., "Invariant Risk Minimization", 2020.

---

### Official Review · Reviewer_Q9e8 · 2025-10-30

**Soundness:** 2
**Presentation:** 2
**Contribution:** 2
**Rating:** 4
**Confidence:** 2

**Summary:**

The paper proposes a framework to improve out-of-distribution generalization in prototype-based routing for LLM adapter selection. During prototype learning, the target task’s adapter is masked to prevent the router from learning noisy features. The authors theoretically show that noisy subspaces cause in- and out-of-distribution samples to become indistinguishable. Experimental results demonstrate that the proposed framework improves the performance of two prototype-based routing methods.

**Strengths:**

- The paper provides a solid theoretical analysis, including proofs, to explain the inadequacy of existing methods.
- The proposed method is conceptually simple and easy to implement.

**Weaknesses:**

- The paper assumes that the entropy difference between in- and out-of-distribution samples adequately captures OOD generalization. However, this assumption overlooks other well-established metrics that can better capture distributional separability, such as the energy score [1].
- The theoretical analysis is not clearly presented. For example, the meaning of the equation in Definition 1 is unclear, and its connection to the rest of the analysis is not well explained. As another example, it is unclear why theorem one indicates that "ensuring that the routing plan induces a low source and high target entropy is a necessary condition for generalization".
-  There is a gap between the theoretically claims and the empirical method. It is unclear why masking the target task's adapter leads a less noisy representation. Some theoretical or empirical analysis is expected.
- The experimental evaluation is not robust and comprehensive.
  - Only the overall task performance is reported. Routing or OOD detection performance is missing.
  - The reported improvements over the baselines are marginal, and the absence of error bars or confidence intervals makes it difficult to assess statistical significance.
- The approach is narrowly tailored to the prototype-based adapter selection setting, which limits its general applicability.

[1] Liu, Weitang, et al. "Energy-based out-of-distribution detection." Advances in neural information processing systems 33 (2020): 21464-21475.

**Questions:**

- In Definition 1, is $f$ a classifier or an encoder?
- In the experiments, what examples are considered as ID and what are considered as OOD?

---

> ### Author Response · Authors · 2025-11-21
> **Rebuttal 1/3**
>
> We thank the reviewer for appreciating the simplicity and the theoretical soundness of our paper, as well as enlightening us about alternate evaluation scores (energy-score) and points to strengthen the paper. We address the reviewers' concerns and questions below.
>
> ---
>
> **W1:** Assumption overlooks other well-established metrics that can better capture distributional separability, such as the energy score
>
> **Response:** Thank you for bringing this metric to our attention. To validate the effectiveness of MAPLE in enhancing distributional separability, we measure the difference in energy score after finetuning via MAPLE and their corresponding initialization (Arrow) for the Phi2 model. We empirically observed that MAPLE improves distribution separability by 0.5, reinforcing the validity of MAPLE. We would add these findings to the revised version.
>
> ---
>
> **W3:** It is unclear why masking the target task's adapter leads to a less noisy representation. Some theoretical or empirical analysis is expected.
>
> **Response:** To illustrate the importance of masking, we consider 2 cases, with masking(MAPLE) and without masking.
> Consider that we have $k$ pretrained LoRAs trained on their corresponding tasks; these LoRAs would have captured some dataset-specific artifacts and other noisy attributes that hinder generalisation. Now we consider 2 cases, with masking (MAPLE) and without masking.
>
> **Case 1:** In the absence of any masking, for an input from task $i$, the top-$k$ activated LoRAs would contain LoRA $i$, since LoRA $i$ was trained on task $i$. Invoking LoRA $i$ for task $i$ would result in low loss during training and consequently weaker gradients, which leads to poor noise removal. Furthermore, elimination of noisy components would only be possible if the noise contributes to incorrect routing, and thus, gradients would be in a direction that eliminates the noise. But in this scenario, the noise in the LoRA $i$ prototypes matches with noise in task $i$ data and thus does not aid in eliminating noise.
>
> **Case2:** Upon masking the LoRA $i$, the remaining LoRAs are used to generate  output for task $i$. Now, since the noise contained in these LoRA prototypes do not align with the noise present in task $i$, they cause erroneous routing, consequently producing higher loss and stronger gradient. Since any noise negatively affects routing, the gradients would be aligned in a direction that eliminates this noise. Thus, on OOD tasks, testing loss would be lower for MAPLE compared to the no masking case, demonstrating superior removal of noise.
>
> **Empirical evidence:**
> The newly added Figure 6 (Appendix E) illustrates this mechanism. With masking, the loss on ID data is significantly higher during training than in the no-masking case, consistent with our explanation that masking produces stronger gradients. In contrast, on OOD data, MAPLE achieves consistently lower loss, showing that it has learned cleaner and more task-agnostic representations. This improvement in loss aligns with the performance gains reported in Table 2 (main text), where masking yields higher accuracy than the no-masking counterpart.
>
> ---
>
> **W4:** The experimental evaluation is not robust and comprehensive
> - **W4a:**   Only the overall task performance is reported. Routing or OOD detection performance is missing.
> - **Response:** We would like to bring the attention of the reviewer to Fig. 4 in the main text, which plots the ID and OOD routing entropy with MAPLE, which provides additional validation of our results. We have illustrated it with a plot rather than exact values in a table, to provide an easy comparison with Fig., which was the motivating observation for MAPLE. Furthermore, since MAPLE implicitly routes the samples to the most relevant experts and doesn't explicitly aim to identify samples as OOD. We will make this point clearer in the main text to avoid any confusion.
>
> - **W4b:**  The reported improvements over the baselines are marginal, and the absence of error bars or confidence intervals makes it difficult to assess statistical significance.
> - **Response:** We appreciate the reviewers' concerns, and to address this,s we have run MAPLE for 2 additional random seeds on Phi-2 and obtained a mean performance of 66.35 ± 0.12.
>
> ---
>
> **W5:** The approach is narrowly tailored to the prototype-based adapter selection setting, which limits its general applicability.
>
> **Response:** Routing methods can also be categorised as non-parametric and parametric methods depending on how the routing function is characterised. The prototype-based routing falls into the category of non-parametric methods; parametric methods obtain routing scores as the output of a neural network parametrised by its weights. MAPLE can be applied to these methods as well by simply zeroing out the output neuron corresponding to the target LoRA, thereby eliminating the corresponding LoRA from the training process.

---

> ### Author Response · Authors · 2025-11-21
> **Rebuttal 2/3**
>
> **Q1:** In Definition 1, f is a classifier or an encoder?
>
> **Response:** In definition 1, $f$ refers to an encoder whose output corresponds to routing scores of LoRAs in the target pool.
>
> ---
>
> **Q2:**  In the experiments, what examples are considered as ID and what are considered as OOD?
>
> **Response:** In our experiments, we constructed the LoRA pool by training LoRAs on 256 tasks from FlanV2 datasets, thus any samples from these tasks are considered as ID samples, while for evaluation,n we considered datasets like Arc-Easy, Arc-challenge, HumanEval, etc. Samples from these tasks are taken as OOD.
>
> ---
>
> **W2:** The theoretical analysis is not clearly presented.
>
> **Response:**
>
> We apologise for the confusion caused regarding the equation in Definition 1, which, we believe, arises from the generic nature of the formulation. We also thank the reviewer for pointing out a subtle error on our part, as well as the need for a clearer explanation of the claimed connection between Theorem 1 and generalization. In addition to addressing these specific issues below, we try to provide a cohesive integration of the theory and the larger problem (as also requested by Reviewer HVHh), in our response to Reviewer HVHh, which we hope improves the overall clarity. We are happy to take further suggestions and incorporate them into our paper.
>
> **Summary:**
> - The equation in Definition 1 characterizes the basis induced by LoRA prototypes as "complete" iff it can be leveraged to route all samples from a distribution such that their labels can be correctly predicted. Our goal thus becomes to learn a basis that is complete according to this definition.
> - Theorem 1 is essentially based on the fact that if we cannot make any distributional assumptions (i.e., OOD samples), no learner, on average, can do any better than the one that is optimal on average(Wolpert & Macready, 1997; [a]). We reformulate this idea and adapt it to our problem setting, where we prove that in OOD settings, such an optimal learner, which exhibits a higher effective rank as a consequence of equally leveraging all experts (optimality due to the inability to make distributional assumptions), is associated with a high target routing entropy. On the other hand, low source entropy is the result of having access to distributional information during training (ID samples), through which a precise subset of experts can be trained in an optimal way, circumventing the need to leverage all experts equally. We provide a detailed version of this answer below.

---

> ### Author Response · Authors · 2025-11-21
> **Rebuttal 3/3**
>
> **Full Answer**
>
> **Definition 1 Equation:** Here, we try to make it more concrete and illustrate how the generic/abstract objects correspond to specific components in our problem.
> The function class $\mathcal{F}$ refers to the family of functions that the router comes from, which, in this case, is a dot-product over learnable LoRA prototypes. The basis $\mathcal{B}$ is essentially the basis for the function class $\mathcal{F}$, which in our case, is the set of LoRA prototypes themselves. $\mathcal{B}$ is used to generate the desired functions in $\mathcal{F}$, which in our case, are the routing coefficients. The equation in Definition 1 can thus be deconstructed into the following parts:
>
> - $\forall(x, y) \in \mathcal{D}$: Refers to all points / samples $x$ with label $y$ in the distribution $\mathcal{D}$.
> - $\exists f \in \mathcal{F}, f(x) \in \operatorname{span}(B)$: Refers to the existence of a function $f$ in the function class $\mathcal{F}$ (that can be derived by linearly combining the vectors in $\mathcal{B}$).
> - $g_{\theta+\phi(f(x))}(x) = y$: Refers to the desired condition that $f$ must satisfy. Specifically, when combined with $g_\theta$ and the LoRA prototypes $\phi$, $f$ should be able to correctly predict the label $y$ for every input $x$. The process of computing $g_{\theta+\phi(f(x))}(x)$ can be found in Figure 2.
>
> In summary, the equation characterizes a basis comprising LoRA prototypes as "complete" iff its constituent vectors can be used to generate routing coefficients such that all samples belonging to a certain distribution can be correctly classified. Therefore, our goal is to learn bases that are complete according to this definition. Throughout the rest of our theoretical analysis, we derive a condition, namely restrictivity (Theorem 1), which can be used to tell apart complete bases from incomplete ones (for instance, those from existing methods), and another condition, namely noisiness (Theorem 2), which prevents the learning of complete bases as a result of such restrictions. The insights derived therefrom are leveraged to design our algorithm, MAPLE.
>
> **Theorem 1 and generalization:** What we intended to mention in the statement is actually the following: "ensuring that the routing plan induces a *higher effective rank of the LoRA prototype space implying* a low source and high target entropy is a necessary condition for generalization". We also validate this claim empirically. By measuring the effective rank of the prototype space before and after training with MAPLE, we observe an increase from 7.35 to 8.47. This rise in effective rank is consistent with the theoretical expectation that MAPLE encourages richer, less degenerate representations, thereby supporting better generalization.
>
> Now, we address the need for having low source and high target routing entropies for better generalization.
>
> - Similar source and target entropies - It means that expert combinations that apply to the source distributions are the only ones that are being utilized for the target. In other words, target-specific features are not being taken into account, implying that the LoRA prototype space does not take into account the distribution shift occurring in the target.
>
> - Low source and high target entropies - While the existence of source-specific experts would naturally lead to a low-source entropy, a high target entropy implies that all experts are being equally leveraged for out-of-distribution target data.
>
> - High source and low target entropies - Not possible with any learning algorithm based on empirical risk minimization (ERM), since the source (train) is the primary distribution that is being learned, and by definition, should have the lowest entropy among all distributions.
>
> Therefore, since we cannot make any distributional assumptions about the target (as it is OOD), according to the No Free Lunch theorem (Wolpert & Macready, 1997), no other combination of experts can generalize on average (across all distributions), any better than the one where we leverage the capabilities of all experts equally. It means that the desirable routing plan should have a low source and high target routing entropy (second point).
>
> From Theorem 1, we know that ensuring a higher effective rank of the LoRA prototype space is one way to achieve the above condition, which is ultimately what we intended to state through our original statement.
>
> References:\
> [a] Gulrajani, "In Search of Lost Domain Generalization", ICLR 2021.

---

> > ### Comment · Reviewer_Q9e8 · 2025-11-24
> >
> > Thank you for the detailed response. One more question: Could you clarify how improving ID and OOD separability in entropy ultimately leads to end-to-end performance gains? To make the theoretical part complete, I would expect a main theorem that connects the two.

---

> > > ### Author Response · Authors · 2025-12-01
> > > **Reply**
> > >
> > > We are glad that almost all the concerns of the reviewer were resolved. We address the reviewers' remaining concern below:
> > >
> > > **R1:** Could you clarify how improving ID and OOD separability in entropy ultimately leads to end-to-end performance gains?
> > >
> > > **Response:** LoRA prototype bases that exhibit a larger gap between ID and OOD entropies are less likely to encode spurious associations arising from data noise, training artifacts, etc. As a consequence, these basis can represent a larger spectrum of distributions. This increases its chances of being complete (as per Definition 1) for a larger spectrum of distributions. This ultimately implies better generalization(performance gains) since the output can only be correctly predicted if it can be completely represented. We have added this clarification in Appendix F.
> > >
> > > **R2:** To make the theoretical part complete, I would expect a main theorem that connects the two.
> > >
> > >  **Response:** Following the reviewers' suggestion, we have added a theorem (Theorem 3) in Appendix F that proves that increased ID and OOD separability in entropy leads to performance gains.

---

### Official Review · Reviewer_HVHh · 2025-10-31

**Soundness:** 2
**Presentation:** 3
**Contribution:** 2
**Rating:** 4
**Confidence:** 2

**Summary:**

This paper introduces MAPLE (Masked Adapter Prototype LEarning), a method to improve OOD generalization when combining multiple LoRA adapters. It is motivated by observing that the current prototype-based routing strategies often yield similar routing entropy for in-distribution and OOD samples, implying poor distinction between the two. The authors provide a theoretical analysis showing that shared noisy subspaces in LoRA prototypes lead to this problem. MAPLE mitigates it by masking the target adapter during prototype learning, forcing the model to refine prototypes that rely on less noisy signals. Empirical results performance gains (~0.5%) over baselines.

**Strengths:**

* Clear motivation, backing theory, and straight-forward mitigation algorithm. The paper connects empirical entropy observations to a formal theoretical framework explaining why simple approaches fail.
* Simple and practical. MAPLE’s masking approach is straightforward, easy to integrate, and doesn’t require modifying base LLMs or LoRAs.

**Weaknesses:**

* The empirical gains appear modest. Compared to other baselines, the improvements are very small, raising the question of whether they might simply result from differences in their initializations?
* The theoretical analysis feels somewhat loose and not fully integrated into a cohesive understanding of the problem.
* The observations seem restricted to text data, limiting the generality of the findings.

**Questions:**

* Figure 1 appears overly stylized and may not reflect real data. Could the authors share the raw observations and clarify how the plot was generated?
* How sensitive is MAPLE to the selection of top-k?
* Given the relatively small absolute gains, is it possible that MAPLE’s improvements stem from random initialization?

---

> ### Author Response · Authors · 2025-11-21
> **Rebuttal 1/2**
>
> We thank the reviewer for appreciating the strengths of our work and providing constructive feedback. We address the reviewer’s concerns and questions below:
>
> **Q1:** Could the authors share the raw observations and clarify how the plot(fig1) was generated?
>
> **Response:** To generate Fig. 1, we considered 5 in-domain (ID) tasks (LoRAs trained on these tasks exist in the LoRA pool) and 5 out-of-domain (OOD) tasks (LoRAs trained on these tasks do not exist in the LoRA pool).
>
> For each task and each layer, we obtain the routing scores and compute the routing entropy using Eq. (1). Concretely, given the logits for the experts at a layer, we construct a categorical distribution and compute its entropy as follows:
>
> ```python
> routing_dist = torch.distributions.Categorical(logits=routing_scores)
> entropy = routing_dist.entropy()
> ```
> The entropy is then averaged across layers, tokens, and samples to obtain a single average entropy value for each task.
>
> The resulting average routing entropies for the 5 ID and 5 OOD tasks used in Fig. 1 are:
>
> ```python
> arrow_id  = [1.36, 1.37, 1.372, 1.29, 1.27]
> arrow_ood = [1.33, 1.32, 1.31, 1.32, 1.36]
> phat_id   = [1.12, 1.04, 1.22, 1.10, 1.12]
> phat_ood  = [1.21, 1.15, 1.06, 1.15, 1.11]
> ```
>
> We visualize these values using a violin plot implemented with `matplotlib.axes.Axes.violinplot`. The plotting code used to produce Fig. 1 is:
>
> ```python
> groups = [arrow_id, arrow_ood, phat_id, phat_ood]
> labels = ["Arrow\n(ID)", "Arrow\n(OOD)", "PHATGOOSE\n(ID)", "PHATGOOSE\n(OOD)"]
>
> fig, ax = plt.subplots(figsize=(fig_w, fig_h))
>
> parts = ax.violinplot(
>     groups,
>     showmeans=True,
>     showmedians=False,
>     showextrema=False,
>     widths=0.8,
> )
>
> ax.set_xticks(range(1, len(labels) + 1))
> ax.set_xticklabels(labels)
> ```
> -----
> **Q2:** How sensitive is MAPLE to the selection of top-$k$?
>
>
> **Response:**  To demonstrate the sensitivity of MAPLE to top-$k$, we report the results of MAPLE with three different values of $k$ ($k$=2,3,4) below:
>
> | Method | k = 2 | k = 3 | k = 4 |
> |--------|-------|-------|-------|
> | Maple  | 66.20 | 66.52 | 66.41 |
>
>  As observed, the performance varies with $k$, and choosing an optimal $k$ further enhances the gains of MAPLE.
>
>  -----
> **Q3/W1:** Given the relatively small absolute gains, Is it possible that MAPLE’s improvements stem from random initialization?
>
>  **Response:** MAPLE’s performance gains are not artifacts of random initialization. Its improvements come from denoising the LoRA representations, which leads to more accurate expert routing and thus better downstream performance. For the results demonstrating MAPLE’s effectiveness (Table 1), the prototypes are initialized using the LoRA representations produced by the corresponding baseline methods (Arrow initialization for Arrow + MAPLE, and PHATGOOSE initialization for PHATGOOSE + MAPLE). To randomize other sources of randomness, such as optimizer states and seeds. We run MAPLE for two additional seeds (totaling 3 runs) with a mean performance of 66.35 ± 0.12, further supporting our claim.
>
>  Since the reviewers' concerns arise from the relatively small absolute gains, we have now added additional measures, such as the validation and testing loss of MAPLE in Fig. 5 of Appendix E. As observed in Fig. 5, the losses exhibit sharper variations with both validation(ID) and testing (OOD) loss decreasing significantly compared to the initialization(Arrow initialization), offering further evidence that the effect is real and not a consequence of random variations.
>
>   -----
>
> **W3:**   The observations seem restricted to text data, limiting the generality of the findings.
>
> **Response:** Our focus on textual data follows the majority of prior work addressing the same problem (e.g., Arrow, PHATGOOSE, LoRAHub etc.), all of which primarily evaluate on language benchmarks. Extending MAPLE to other modalities, such as vision, is conceptually straightforward but would require non-trivial adjustments to our current implementation and training pipeline (e.g., modality-specific backbones and adapter architectures). In principle, MAPLE can be applied to the vision domain by using a pool of pretrained LoRAs and masking the corresponding target-task LoRA during finetuning. We agree that evaluating MAPLE beyond text is an interesting and valuable direction, and we will highlight this as a promising avenue for future work.
>
>   -----

---

> ### Author Response · Authors · 2025-11-21
> **Rebuttal 2/2**
>
> **W2:** The theoretical analysis feels somewhat loose and not fully integrated into a cohesive understanding of the problem.
>
> **Response:** We thank the reviewer for bringing to our notice the lack of cohesion between the theory and the overall problem. We fully recognize this issue and aim to address it through this rebuttal, to the best of our abilities. We are happy to receive any feedback and enhance our exposition even further, if need be.
>
> **Summary:** We view routing as an embedding of the input into a vector space, of which, the constituent bases are the LoRA prototype vectors. We show, via *Theorem 1*, that when the effective rank of this vector space is low, the model is forced to use similar routing plans for both ID and OOD samples, hindering OOD-specific adaptation.
> *Theorem 2* then shows that the components of the LoRA prototype space that restrict its effective rank / dimensionality (leading to the above hindrance) must be noisy, for which we provide direct empirical evidence in this rebuttal.
>
> **Full Answer:**
>
> We view routing as an embedding of the input into a vector space whose constituent bases are the LoRA prototype vectors. We show, via Theorem 1, that when the rank of this vector space is low, one is likely to observe similar source and target routing entropies. The intuition behind this is that a low rank prototype basis "restricts" the span of the vector space in which the input could be embedded, meaning that the model is forced to use similar routing plans for both ID and OOD samples, hindering OOD-specific adaptation.
>
> The fact that existing methods exhibit similar source and target entropies is illustrated empirically in Figure 1. To provide empirical evidence to the premise that such overlap in entropy occurs due to low-rank LoRA prototypes, we conduct further experiments as part of this rebuttal and show that the rank of the LoRA prototype basis (rank = 7.35) is lower for existing methods as compared to our proposed MAPLE (rank =8.47). Furthermore, since Theorem 1 equivalently predicts an increase in rank to lead to low source and high target entropy, we conduct the experiment shown in Figure 4, where such a phenomenon is indeed observed. Source entropy is significantly lower than the target for MAPLE, which is seen to yield higher rank prototypes based on our recent experiments. The fact that LoRA prototypes become more orthogonal (implying higher rank) with MAPLE fine-tuning can also be observed in Figure 3, where the cosine similarity between different prototypes is seen to generally decrease, justifying the applicability of Theorem 1.
>
> Following from the above, Theorem 2 attempts to characterize what exactly is responsible for the restrictivity (as described in Theorem 1) of the LoRA prototype space. It concludes that the components of the LoRA prototype space that cause restrictivity must be noisy, i.e., irrelevant to all distributions. Although this claim is indirectly validated by our improved empirical OOD gains, we aim to provide more direct evidence for it. To this end, we performed the following experiment for this rebuttal:
>
> Consider the initial (pre-finetuning) prototype vector to be $A$ and the post-finetuning prototype vector to be $B$. As $A$ gets closer to $B$, the difference vector $(A-B)$, which corresponds to the subspace being removed from $A$ in order to reach $B$, gets closer to $B$ too. As a result, $B \cdot (A-B)$ increases. Therefore, the magnitude of the dot-product of $(A-B)$ with any vector $X$, i.e., $X \cdot (A-B)$, can be used as a measure of the relevance (or inverse measure of noisiness) of the removed subspace $(A - B)$ to that of $X$, which reaches its maximum when both $(A - B)$ and $X$ are equal to $B$.
>
> In summary, the difference vector $(A - B)$ can be thought of as the subspace removed under denoising and consequently, the dot product $X \cdot (A - B)$ can be used as an inverse measure of the noisiness of the eliminated subspace with respect to $X$. The lower the dot-product, the more noisy the removed subspace, and vice versa.
>
> We leverage this idea to estimate the degree of denoising achieved by our method. To do so, we compute the dot product of the difference vector (pre-finetuning minus post-finetuning(denoted as Δ)) with each of the source and target distributions. We then compare these values with the dot products between the inputs and the MAPLE prototypes. As shown below, the dot products involving the difference vector are markedly smaller:
>
> | Prototype | $(a \cdot b)$ |
> |-----------|---------------|
> | MAPLE     | 0.272         |
> | Δ         | 0.136         |
>
>
> This implies the noisiness of the subspaces eliminated through fine-tuning, relative to the actual prototype bases that our proposed MAPLE converges to.
>
>   -----

---

> > ### Comment · Reviewer_HVHh · 2025-11-27
> >
> > I thank the authors for their detailed response. While some of my concerns were addressed, others remain. Consequently, I will maintain my current score.
> >
> > **R1** Thank you for providing the technical details. My original query aimed to understand the underlying data, as the plot appeared oversimplified. It seems this simplicity stems from the small number of LoRAs and tasks used. Although this analysis was conducted to demonstrate motivation, the generalizability of these observations appears limited because they are based on only 10 tasks and a single model.
> >
> > **R2** Thank you for providing the additional observations.
> >
> > **R3** Regarding the result of 66.35 ± 0.12: The baseline number (66.26) falls within one standard deviation of this result (66.35 - 66.26 = 0.09). Therefore, the authors' claim regarding the statistical significance of this improvement is unconvincing.
> >
> > **R4** I appreciate the authors' interest in expanding this work to multi-modal tasks.
> >
> > **R5** Thank you for the detailed discussion. However, I feel the authors' interpretation of the routing mechanism in the context of LoRA adapters is somewhat oversimplified. Specifically, LoRA adapters do not induce a purely linear transformation of the network's behavior, which complicates this perspective.

---

> > > ### Author Response · Authors · 2025-12-02
> > > **Response**
> > >
> > > We thank the reviewer for the continued engagement during the discussion phase. We are glad that many of the earlier concerns have been addressed, and we respond here to the remaining points below.
> > >
> > > **R1** Although this analysis was conducted to demonstrate motivation, the generalizability of these observations appears limited because they are based on only 10 tasks and a single model.
> > >
> > > **Response.** To address this concern, we extended our analysis to a broader setting. Specifically, we now compute routing entropy on 30 tasks (15 in-distribution and 15 out-of-distribution) across two models: Phi-2 and Llama 3. We report the averaged statistics below:
> > >
> > > | Model   | ID routing entropy | OOD routing entropy |
> > > |---------|--------------------|---------------------|
> > > | Phi-2   |     1.34(0.12)     |      1.35 (0.01)    |
> > > | Llama 3 |     1.29(0.15      |      1.30 (0.09)    |
> > >
> > >
> > > Across this expanded set of tasks and models, we observe the same qualitative trend as in our original experiments: the relationship between routing entropy and OOD behavior is preserved. This suggests that our observations are not an artifact of undersampling or tied to a single model family, but instead reflect a more general phenomenon.
> > >
> > > **R3** Regarding the result of 66.35 ± 0.12: The baseline number (66.26) falls within one standard deviation of this result (66.35 − 66.26 = 0.09). Therefore, the authors’ claim regarding the statistical significance of this improvement is unconvincing.
> > >
> > >
> > > **Response.** We believe that because of the lack of details in our rebuttal, the reviewer might have misunderstood our earlier response. We apologise for this oversight and clarify the concerns of the reviewer in this regard. The result reported by us, 66.35 ± 0.12, corresponds to the setting of  Arrow + MAPLE(Phi2); the corresponding baseline for this would be Arrow(Phi2), which has a mean of 65.94 and thus lies outside the standard deviation. We would also like to clarify that the additional plots that we added also compare these 2 settings (Arrow and Arrow+MAPLE) for Phi2.
> > >
> > > **R5** However, I feel the authors' interpretation of the routing mechanism in the context of LoRA adapters is somewhat oversimplified. Specifically, LoRA adapters do not induce a purely linear transformation of the network's behavior, which complicates this perspective.
> > >
> > > **Response.** We would like to clairfy that we donot make the assumption "LoRA adapters induce a purely linear transformation of the network's behavior". We considered the router which is linear and utilize its linearity property for all the analysis without making any additional assumptions. We have made this distinction clearer in the main text.

---

### Author Response · Authors · 2025-12-02
**Rebuttal Summary**

Dear Area Chair,

We appreciate your additional effort in handling this unforeseen situation. To support your assessment, we briefly summarize the current status of our submission below.

**Reviewer HVHh** (rating: 4) had most of their concerns resolved during the discussion phase, with only three remaining points at the time of the abrupt interruption. We have now addressed these remaining concerns with additional experiments and clarifications.

**Reviewer Q9e8** (rating: 4) also had nearly all concerns resolved during the discussion phase and raised only one further question, which we have now answered in detail.

**Reviewer 8KQz** (rating: 4) was unfortunately unable to engage with us during the discussion phase due to the current circumstances. Nevertheless, we have conducted additional experiments and provided clarifications that address all the concerns raised in their original review.

Thank you again for your time and consideration

---

### Meta-Review · Area_Chair_Hxwy · 2026-01-06

**Summary:**

This paper proposes MAPLE, a refinement of LoRA for fine-tuning LLMs on tasks that require OOD generalization. Despite the simplicity of the proposed method, two reviewers explicitly mentioned that the achieved gains are somewhat modest. Reviewers also raised concerns about the consistency between theory and algorithm, the overall motivation, and the generality of the proposed method, resulting in scores of 4, 4, 4. The authors have acknowledged these concerns and posted a rebuttal. Two out of three reviewers participated in the discussion and maintained the original scores.

From my perspective, the main concern of marginal empirical gains remains after the discussion, and while the authors claimed that most of the concerns have been addressed, the reviewers have actually raised more questions during the rebuttal (e.g., Reviewer HVHh), and I do not think the authors have addressed them convincingly. I thus recommend rejection.

**Reviewer Concerns:**

Reviewers have raised concerns about the marginal empirical gains of MAPLE, the consistency between theory and algorithm, the overall motivation, and the generality of the proposed method. The main concern of marginal empirical gains remains after the discussion, and while the authors claimed that most of the concerns have been addressed, the reviewers have actually raised more questions during the rebuttal (e.g., Reviewer HVHh), and I do not think the authors have addressed them convincingly.

**Reviewer Scores:**

Two out of three reviewers participated in the discussion and maintained the original score. One reviewer did not participate in the discussion, and I think their score would not change since the authors have not provided a clear and convincing response to some of their questions, e.g., regarding to the OOD generalization problem formulation.

---

### Decision · Program_Chairs · 2026-01-26

Reject